# Kinematic and Dynamic Scaling of Copepod Swimming

**Leonid Svetlichny [1],\*** , **Poul S. Larsen [2]** and **Thomas Kiørboe [3]**

1    I.I. Schmalhausen Institute of Zoology, National Academy of Sciences of Ukraine, Str. B. Khmelnytskogo, 15, 01030 Kyiv, Ukraine
2    DTU Mechanical Engineering, Fluid Mechanics, Technical University of Denmark, Building 403, DK-2800 Kgs. Lyngby, Denmark; psl@mek.dtu.dk
3    Centre for Ocean Life, Danish Technical University, DTU Aqua, Building 202, DK-2800 Kgs. Lyngby, Denmark; tk@aqua.dtu.dk
\*    Correspondence: leonid.svetlichny@gmail.com

**Abstract:** Calanoid copepods have two swimming gaits, namely cruise swimming that is propelled by the beating of the cephalic feeding appendages and short-lasting jumps that are propelled by the power strokes of the four or five pairs of thoracal swimming legs. The latter may be 100 times faster than the former, and the required forces and power production are consequently much larger. Here, we estimated the magnitude and size scaling of swimming speed, leg beat frequency, forces, power requirements, and energetics of these two propulsion modes. We used data from the literature together with new data to estimate forces by two different approaches in 37 species of calanoid copepods: the direct measurement of forces produced by copepods attached to a tensiometer and the indirect estimation of forces from swimming speed or acceleration in combination with experimentally estimated drag coefficients. Depending on the approach, we found that the propulsive forces, both for cruise swimming and escape jumps, scaled with prosome length ($L$) to a power between 2 and 3. We further found that power requirements scales for both type of swimming as $L^3$. Finally, we found that the cost of transportation (i.e., calories per unit body mass and distance transported) was higher for swimming-by-jumping than for cruise swimming by a factor of 7 for large copepods but only a factor of 3 for small ones. This may explain why only small cyclopoid copepods can afford this hydrodynamically stealthy transportation mode as their routine, while large copepods are cruise swimmers.

**Keywords:** copepods; cruising; escape swimming; kinematics; hydrodynamics; power; cost of transport

## 1. Introduction

The swimming of pelagic copepods is based on the principle of rowing strokes with oar-like limbs. The anatomy of the body structure is directly related to the way of swimming, and copepods are divided into two main groups: the ancient Gymnoplea and the more recent Podoplea [1,2]. In Gymnoplea, which includes the Calanoida, both the cephalic and thoracic limbs participate in propulsion. The cephalic appendages perform the combined functions of feeding and steady cruise swimming [3]. In Podoplea, only the thoracic limbs—the swimming legs—are involved in swimming. The thoracic limbs in all copepods, with the exception of some parasitic taxa, are used for jumping.

The first descriptions of the kinematics of the cephalic appendages of copepods belonged to Storch and Pfisterer [4] and Cannon [3]. Subsequently, they were supplemented by Lowndes [5] and developed by Gauld [6] and Petipa [7]. The purpose of these experimental works was to elucidate the

copepod feeding mechanisms, and they were performed using filming, polygraphs, and stroboscopic photography. The concept of filtration feeding was developed based on these studies. More advanced high-speed filming later revealed that the feeding mechanism is not the filtering of particles through a sieve; rather, the feeding current is a scanning current [8]. The use of high-speed filming made it possible to reveal new details of the complex interaction of the cephalic limbs during feeding and movement, and it was demonstrated that the frequency of cephalic limb beating in copepods varies between 20 and 40 Hz but can reach 70–80 Hz [9–12]. Thus, even during slow swimming, the limbs oscillate so fast that analyzing their action requires video recordings with a frequency of 700–800 Hz to obtain a good resolution of the leg stroke phase. During escape swimming, the requirements for recording frequency are even higher because limb frequencies may be as high as 200 Hz [13].

Storch [14] may have been the first to use a high-speed movie camera at 120 frames per second to study the jumping behavior of freshwater cyclopoid copepods. He described the metachronal strokes of the thoracic legs of *Cyclops scutifer* during avoidance response. Subsequent studies, using increasingly higher frame rates of up >3000 fps, estimated incredibly high swimming speeds during escape jumps of >500 body lengths per second, and they provided detailed descriptions of the movement of the feeding appendages and swimming legs during cruise swimming and jumps [7,15–22]. These high resolution observations of swimming speeds and appendage kinematics provided the basis for estimations of the force production and energetics of copepod propulsion [23–28]. From observations of speed or acceleration, together with estimates of drag of the moving body or limbs, it is possible to estimate force production.

An alternative approach to estimate force production during swimming and jumping is to directly measure forces of animals tethered to a tensiometer [29–32], a spring [33], or an aluminum wire whose deflection is calibrated and monitored by a displacement sensor [34,35].

The aim of this synthesis was to describe limb kinematics and examine the magnitudes and size scaling of force production and energy expenditure during cruise and jump swimming in copepods. We combined available literature data with our own new data on swimming speed, appendage kinematics, drag measurements, force measurements on attached specimens, and direct and indirect estimates of force production. We analyzed observations by means of simple theoretical models, and we provide correlations that reflect size scaling laws for kinematics, force, power, and drag. All symbols used are listed in Table 1.

**Table 1.** List of symbols.

| | |
|---|---|
| $a$ | acceleration |
| $c$ | hydrodynamic shape factor |
| $C_d$ | coefficient of drag |
| $C_t$ | energy consumption per unit body mass and time |
| $D$ | diameter of body |
| $D$ | duration |
| $E$ | energy |
| $F$ | frequency of beat |
| $K$ | empirical constant |
| $l_a$ | effective length of second antenna |
| $L$ | prosome length |
| $M$ | body mass |
| $N$ | power, energy per unit time |
| Re | Reynols tal, $\rho L U / \mu$ |
| $R_d$ | drag force |
| $R_p$ | propulsive force |
| $S$ | sectional area of body |
| $S$ | distance |
| $S_{loc}$ | locomotor step length |
| $U$ | body speed |
| $U_a$ | circular speed of second antenna |

**Table 1.** *Cont.*

| Greek | |
|---|---|
| α | angle of second antenna beat |
| μ | dynamic viscosity |
| ν | kinematic viscosity, $\mu/\rho$ |
| ρ | density |

| Subscripts | |
|---|---|
| att | attached, tethered to force sensor |
| cr | cruising, free |
| d | drag |
| esc | escape jump |
| kick | kick, jump |
| max | maximal |
| mean | mean |
| min | minimal |
| p | propulsion |
| st | stroke phase |

## 2. Locomotor Function of Appendages

### 2.1. Cruise Swimming

The cephalic appendages serve the functions of propulsion and the capture of food particles. Depending on the degree to which the cephalic appendages combine these functions, one can identify three main kinematics (Figure 1). For an older group of cruising feeders (Figure 1A), such as *Calanus*, *Paracalanus* and *Pseudocalanus* that consume food particles during continuous uniform swimming, the main feature of their limb movement is the antiphase action of the second antennas and maxillipeds [3,5,10].

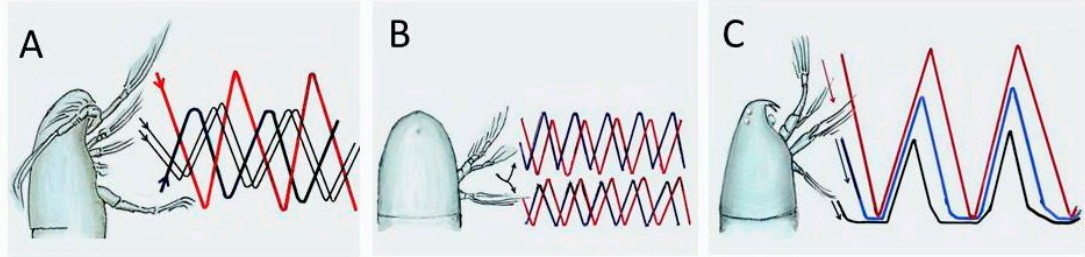

**Figure 1.** Schematic representation of the action of cephalic appendages in calanoid copepods in terms of their angular movements during cruise swimming. Each line starts from the nearest drawn cephalic appendages. (**A**): *Calanus helgolandicus*. The red and two thin black lines directed downward correspond to the angular movement of second antennas, mandibles, and maxillas; the blue line directed up corresponds to the movement of maxillipeds (from [12]). (**B**): *Eurytemora affinis*. The upper red and blue lines show the movement of exopodite and endopodite, respectively, of the second antenna; the lower red and blue lines correspond to the movement of the exopodite and endopodite, respectively, of the mandible. (**C**): *Anomalocera patersoni*. Red, blue, and black lines correspond to movement of second antennas, mandibles, and maxillas, respectively.

The limb kinematics determines the resulting propulsive force, which allows the copepods to swim steadily (Figure 2A). This is evidenced by experiments with the amputation of individual pairs of cephalic appendages. After the amputation of the maxillipeds, the force resulting from the partially antiphase action of the second antennas, mandibles, and maxillas has been found to remain the same, but a pronounced inverse component of the force has been found to appear (Figure 2B). It was found that the amplitude of the force of the second antennae alone is higher again than the force resulting

from the combined action of all the cephalic limbs (Figure 2C). As a consequence, the net propulsion force is reduced with the simultaneous multidirectional action of all limbs.

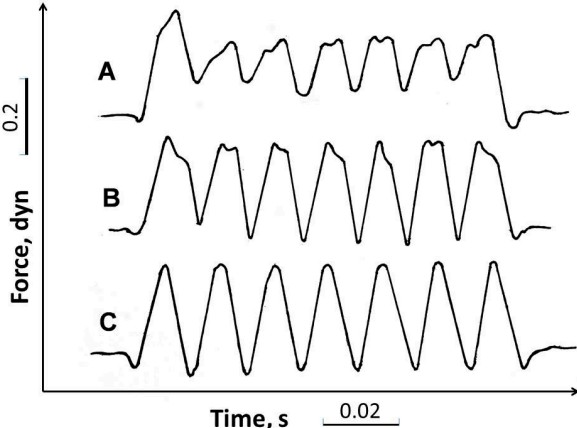

**Figure 2.** Mechanograms of the resulting propulsive force versus time of the cephalic limbs in *Calanus helgolandicus* attached to a semiconductor force sensor. (**A**): All cephalic limbs active. (**B**): After the amputation of maxillipeds. (**C**): After the amputation of all mouth appendages, leaving only the second antennas active (from [10]).

The next group, "feeding current feeders" (Figure 1B), produce a feeding current while (almost) hovering. The group includes *Eurytemora*, *Pseudodiaptomus,* and *Limnocalanus*, which create a constant propulsive force mainly due to the antiphase action of the endo- and exopodites of antennas and mandibles, which Lowndes [5] figuratively compared with "trick-swimming motion." In *Temora longicornis*, the first maxillae participate in the movement too, though they do so with a significantly lower amplitude of action [17]. During feeding, these species can hover in water or attach to a substrate to select food particles from water currents moving along the body. Such calanoid copepods often alternate feeding current feeding with small relocation jumps.

A third group of larger copepods with a predominantly predatory type of feeding, such as *Pontella* and *Anomalocera*, capture food with the maxillipeds and move through the water thanks to the sequential power strokes of the second antennas, mandibles, and maxillas (Figure 1C). During the synchronous return movement of these limbs, the speed of the copepods markedly decreases, making their swimming erratic.

Yet another group can be separated, i.e., copepods that display less regular kinematics for feeding and swimming and belong to the evolutionarily latest ambush feeders. This group includes the Acartiidae family. They can be ambush feeding while slowly sinking and intermittently performing short relocation jumps to remain suspended, or they can perform short feeding bouts similar to cruise feeders, interrupted by periods of passive sinking [11,36].

Unlike calanoid copepods, cyclopoid copepods have completely lost their ability to move by using the cephalic appendages. The cyclopoids are extreme ambush feeders that capture single food objects and move using only the thoracic limbs and the abdomen (see below). This may be a classic example of a progressive reduction in limb function (oligomerization).

*2.2. Jump Swimming*

The jumping, erratic swimming of Gymnoplea and Podoplea is of the same type and is due to the sequential strokes of the thoracic limbs (swimming legs) that have a very similar structure in all free-living copepods [37]. In the Podoplea, this is the routine way of locomotion, and in both groups, the jumps can be particularly powerful—escape jumps—and accelerate the copepod within milliseconds to >500 body lengths per second [19].

The difference between species lies in the number of limbs that generate thrust. For example, all Cyclopoida and some Calanoida-like pontellids have four pairs of swimming legs, while copepods of the genus *Calanus* have five pairs. The thoracic legs rotate because of the contraction of nearly all indirect truncal muscles—both the longitudinal, ribbon-like ones located mainly at the dorsal side along the whole body and the transverse ones located in each thoracic segment (Figure 3A,B). All thoracic limbs of *Calanus helgolandicus* are united in one kinetic chain (Figure 3C) that defines the metachronal sequence of their beating during the power stroke phase of the kick [38]. A similar fastening of the swimming legs has been described for *Rhincalanus* [39] (Figure 80) and *Mormonilla* [40] (Figure 81). The longitudinal dorsal muscles, when contracting, telescopically compress thoracic segments at the dorsal side, and, vice versa, they expand them at the ventral side so that the legs of the last thoracic segment are the first to be kicked into action. Each pair of legs turns under the principle of a lever, whose rotation axis is the place of connection of the intercoxal sclerite with the sternal sclerite located at the front segment. The connections of the coxopodites of the limbs, and the ventral projections of tergite are the points of muscle application.

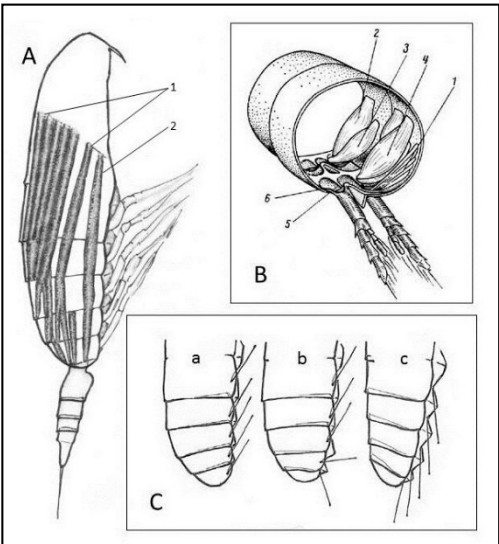

**Figure 3.** *Calanus helgolandicus.* (**A**): Longitudinal truncal muscles scheme: **1**—dorsolateral muscle bundles; **2**—ventral muscle bundles. (**B**): Indirect dorsoventral muscles of the third and fourth pereion segments: **1**—protractors (prepared in the fourth segment); **2**—lateral retractor; **3**—small medial retractor; **4**—big medial retractor (prepared in the fourth segment); **5**—medial apodema; **6**—lateral apodema of thoracic limbs (from [29] with changes). (**C**): Scheme of activity of kinetic chain of limbs: **a**—initial position of limbs; **b**—stroke by 5 and 4 pairs; **c**—position of limbs at the end of the stroke phase (adapted from [38]).

The backward shifting of these points relative to the segment located ahead leads to the leg turning from a forward position to the position initially perpendicular to the body. Subsequently, the contraction of the longitudinal muscles of the body is enhanced by the contraction of dorsoventral muscles compressing the given segment in the transverse plane. As a result, the legs are brought into rear position by the joint efforts of longitudinal and transverse muscles. This mechanism of limb action is similar to the indirect flight mechanism of insects [41]. Thanks to the large number of muscles brought into action during a power stroke, copepods are capable of developing a mechanical muscle power output that is extreme for animals, including flying insects [22,34,42,43].

The abdomen may function as a rudder during jumps [5] and in some species, e.g., *Oithona davisae*, also provides propulsion force (personal observation).

In the calanoid copepods, the first antennae (or antennules) are the longest appendages of the body. When extended, they stabilize the position of the body [44] and slow down the sinking speed of inactive

individuals, since all copepods, except for some phases of their ontogenesis, are negatively buoyant [45]. Some studies have suggested that in *Eurytemora affinis*, the antennae are active contributors to the production of propulsive force [46]. However, numerous high-speed studies of relocation jumps have shown that the antennules are pressed close to the body during the first power stroke of a jump event, and they then remain passive during subsequent power strokes [16,19,30,47]. An exception is the swimming of males of the genus *Oithona* that swim due to power strokes of almost all limbs, including short antennae, cephalic appendages, thoracic limbs, and the abdomen [47,48].

## 3. Scaling of Swimming Kinematics

### 3.1. Cruising of Calanoid Copepods

To identify the large-scale patterns of cruise swimming copepods, we used our own and published data obtained by high-speed methods to simultaneously determine the swimming speed ($U$, cm s$^{-1}$) and beat frequency of cephalic appendages ($F$, Hz) as a function of the body length ($L$, cm) of individual specimens (Table 2). Swimming speed increases with body length to a power of approximately 1.4; 'the locomotor step length' ($S_{loc}$), i.e., the distance that the copepod covers during one beat cycle, increases approximately with the square of the body length; and limb beat frequency varies approximately inversely with the square root of body length (Figure 4).

**Table 2.** Kinematic parameters of cruise swimming calanoid copepods at 20 °C. $L_{pr}$: prosome length; $L_{an}$: effective length of second antenna, measured as the distance from body to mid-area of marginal bristles of endopodites; $n_{ind}$: number of individuals; $n_m$: number of measurements; $F$: frequency of cephalic appendages at cruising speed; $U$: horizontal body speed; $S_{loc} = U_{body}/F$: locomotor step.

| Species | $L$ (cm) | $n_{ind}/n_m$ | $F$ (Hz) | $U$ (cm s$^{-1}$) | $S_{loc}$ (cm) | Source |
|---|---|---|---|---|---|---|
| *Paracalanus parvus* | 0.063 | 8/56 | 63.9 ± 12.4 | 0.31 ± 0.15 | 0.005 | |
| | -"- | 3/8 | 75.9 ± 5.3 | 0.8 ± 0.25 | 0.011 | |
| *Acartia tonsa* | 0.084 | 8/86 | 77.8 ± 4.6 | 0.33 ± 0.4 | 0.004 | |
| | -"- | 4/32 | 66.0 ± 5.1 | 0.4 ± 0.6 | 0.006 | |
| *Centropages ponticus* | 0.084 | 6/34 | 69.0 ± 8 | 0.45 ± 0.13 | 0.007 | |
| *Pseudocalanus elongatus* | 0.084 | 7/24 | 41.8 ± 7.3 | 0.56 ± 0.27 | 0.013 | |
| *Euritemora affinis* | 0.08 | 2/54 | 68.4 ± 3.2 | 0.64 ± 0.29 | 0.009 | |
| | -"- | 1/3 | 66.7 ± 2.4 | 0.45 ± 0.13 | 0.007 | |
| *Centropages typicus* | 0.112 | 11/109 | 39.6 ± 4.1 | 0.81 ± 0.38 | 0.020 | Present data |
| | -"- | 1/4 | 42.7 ± 1.3 | 1.37 ± 0.33 | 0.032 | |
| *Limnocalanus macrurus* | 0.18 | 4/40 | 41.7 ± 5.5 | 0.84 ± 0.09 | 0.020 | |
| | -"- | 1/5 | 39.7 ± 4.3 | 0.53 ± 0.01 | 0.013 | |
| *Pontella mediterranea* | 0.20 | 5/26 | 23.4 ± 1.2 | 3.1 ± 0.57 | 0.132 | |
| | -"- | 2/8 | 26.3 ± 1.7 | 2.5 ± 0.8 | 0.153 | |
| *Calanus helgolandicus* | 0.27 | 7/82 | 36.0 ± 2.7 | 2.16 ± 0.45 | 0.060 | |
| | -"- | 4/9 | 41.3 ± 5.2 | | | |
| *Anomalocera patersoni* | 0.25 | 3/38 | 26.4 ± 10.1 | 3.5 ± 1.7 | 0.133 | |
| | -"- | 6/26 | 21.3 ± 4.1 | | | |

**Table 2.** *Cont.*

| Species | $L$ (cm) | $n_{ind}/n_{m}$ | $F$ (Hz) | $U$ (cm s$^{-1}$) | $S_{loc}$ (cm) | Source |
|---|---|---|---|---|---|---|
| *Pseudodiaptomus marinus* | 0.082 | 5/39 | 80.4 ± 6.8 | 0.24 ± 0.06 | 0.003 | |
| *Paracalanus parvus* | 0.06 | | 63.0 ± 6 | 0.35 ± 0.05 | 0.006 | |
| | 0.063 | | 72.3 ± 4 | | | |
| *Pseudocalanus elongatus* | 0.08 | | 45.2 ± 5 | 0.48 ± 0.17 | 0.011 | |
| *Centropages ponticus* | 0.086 | | 64.0 ± 1 | | | |
| *Acartia clausi* | 0.095 | | 51.7 ± 10 | | | |
| *Pontella mediterranea* | 0.24 | | 27 ± 3 | | | [32] |
| *Calanus helgolandicus* | 0.26 | | 39.1 ± 5 | 2.69 ± 0.1 | 0.068 | |
| *Neocalanus gracilis* | 0.25 | | 28.0 ± 2 | | | |
| *Euchirella messinensis* | 0.35 | | 29.1 | | | |
| *Euchaeta marina* | 0.3 | | 55.0 ± 5 | | | |
| *Pleuromamma abdominalis* | 0.23 | | 37 ± 3 | | | |
| *Phaenna spinifera* | 0.14 | | 59.5 ± 3 | | | |
| *Calanus helgolandicus* | 0.27 | | | 3.2 | | |
| *Rhincalanus nasutus* | 0.5 | | | 0.59 | | |
| *Euchirella curticauda* | 0.36 | | | 2 | | [49] |
| *Euchaeta marina* | 0.33 | | | 2.5 | | |
| *Scolecthrix sp,* | 0.18 | | | 1.1 | | |
| *Anomalocera patersoni* | 0.31 | | | 5.32 | | |
| *Diaptomus kenai* | 0.18 | | | 0.5 ± 0.1 | | |
| *Diaptomus tyrelli* | 0.08 | | | 0.05 | | [50] |
| *Diaptomus hesperus* | 0.15 | | 50 | 0.31 | 0.006 | |
| *Eucalanus pileatus* | 0.14 | | 18 | | | [51] |
| *Paracalanus parvus* | 0.07 | | 83 | | | |
| *Centropages typicus* | 0.14 | | 55 | | | [52] |
| *Calanus sinicus* | 0.23 | | | 1.14 | | [53] |
| *Temora longocornis* | 0.09 | | 32. ± 3 | | | [54] |
| *Eurytemora hirundoides* | 0.084 | | | 0.34 | | [55] |
| *Acartia granii (females)* | 0.101 | | | 0.33 ± 0.5 | | |
| *Temora longicornis (females)* | 0.074 | | | 0.14 ± 0.19 | | |
| *Temora stylifera (females)* | 0.107 | | | 0.33 ± 0.35 | | |
| *Pseudocalanus elongatus (females)* | 0.079 | | | 0.2 ± 0.26 | | |
| *Acartia granii (males)* | 0.088 | | | 0.34 ± 0.84 | | [48] |
| *Temora longicornis (males)* | 0.068 | | | 0.3 ± 0.23 | | |
| *Temora stylifera (males)* | 0.099 | | | 0.72 ± 0.46 | | |
| *Pseudocalanus elongatus (males)* | 0.064 | | | 0.28 ± 0.3 | | |
| *Temora longicornis* | 0.085 | | 40.7 ± 8 | 0.48 ± 0.9 | | [17] |
| *Centropages velificatus* | 0.12 | | | 0.7 | | [56] |
| *Paracalanus aculeatus* | 0.1 | | | 0.2 | | |
| *Euchaeta rimana* | 0.25 | | | 0.75 ± 0.04 | | [57] |

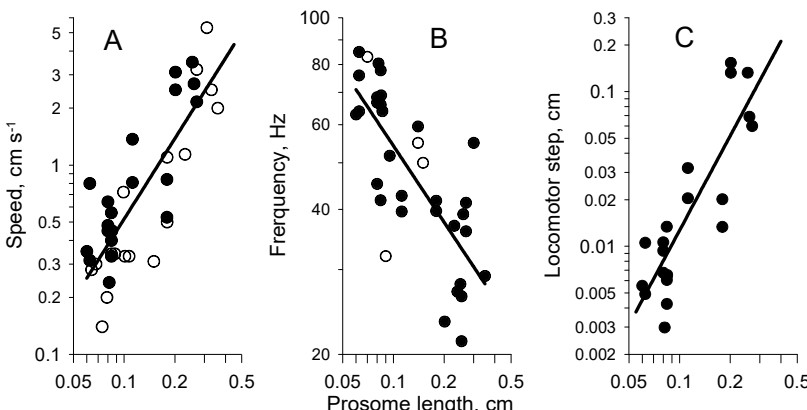

**Figure 4.** (**A**): Regressions of average speed (*U*). (**B**): Limb beat frequency (*F*). (**C**): 'Locomotor step length' ($S_{loc}$) versus prosome length (*L*) during cruise swimming (data from Table 2). Black circles are own data obtained from 1200 fps videos. Empty circles are literature data. The power-law regressions were based on all data, $U = 13.4\ L^{1.4}$ ($R^2 = 0.69$); $F = 16.0\ L^{-0.53}$ ($R^2 = 0.59$); $S_{loc} = 1.36\ L^{2.03}$ ($R^2 = 0.73$).

### 3.2. Kinematic Analysis of Escape Reaction

Since even modern high-speed cameras do not allow for long-term recordings of animal activity, the copepod escape reaction may be synchronized with video records by various external means of stimulation, such as short, weak electrical pulses [30,31] or photic and hydrodynamic stimuli [19,20,58]. In our studies, we used short electrical impulses (see [30,31]). With this dosage, we observed a stable and maximum motor response. Another advantage is that all the copepod species studied by us showed positive galvanotaxis. With the lateral placement of the electrodes, this increased the likelihood of individuals moving in the focal plane of the camera lens, therefore providing sharper images. After each period of stimulation, the copepods were replaced with new animals.

Video sequences showing specimens moving in the focal plane were selected for frame-by-frame analysis. We digitized the geometric center of the prosome of the copepod and computed velocities from the change of this position between frames. Video recording was performed at 1200 fps with a back collimated beam of light from a 5 W LED lamp. All measured parameters describing the kinematics of the escape reaction are explained in the Supplementary Table S1.

It has been previously shown that the direction of trajectory can change dramatically, even up to a complete turn, during a power stroke [19,28].

However, even during rectilinear movement, power strokes by the abdomen and swimming legs cause a dorsal rotation of the body, while returning the limbs to their original position leads to the rotation of the body in the ventral direction [31]. Particularly pronounced are such body rotations in copepods with elongated abdomens. For example, in the cyclopoid copepod *Oithona davisae* with a total body length of 0.05 cm, the ventral deviation of the body axis from the direction of movement at the end of the kick has been found to reach 90°. From this position, the next kick starts (Figure 5).

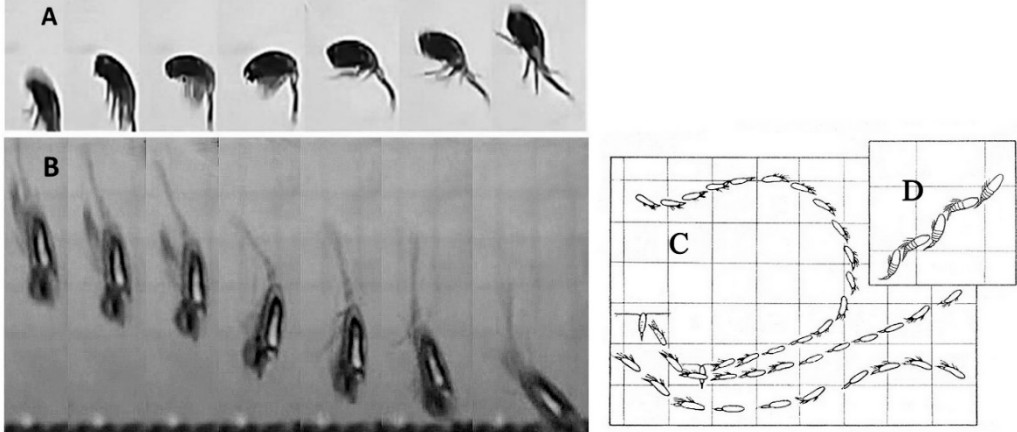

**Figure 5.** Instantaneous body positions of *Oithona davisae* (**A**) and *Limnocalanus macrurus* (**B**) during escape kick, the trajectory of three individuals of *Calanus helgolandicus* stimulated by electrical impulses (**C**), and the instantaneous positions of the body at the end of stroke and recovery phases of kick (**D**). Note, only characteristic body positions are shown in (**A**,**B**) (present data). (**C**,**D**) are from [31].

In larger (0.3 cm) calanoid copepods of *Limnocalanus macrurus* with a very long abdomen, the turning of the body axis has been found to reach 45°. For copepods with a relatively short abdomen, such as *Paracalanus* and *Calanus*, the angular amplitude of oscillations of the body axis relative to the direction of motion is about 30°. Nevertheless, in small *Acartia tonsa* (<0.1 cm), the body angle can vary within 55° [13]. All copepods also rotate their body around their longitudinal axes [19,31].

3.2.1. Instantaneous and Average Speed of Escape Reaction

A complete escape reaction is made up of a series of kicks [22,31,59,60]. During the inertial phase between kicks, the velocity decreases to $U_{min}$ immediately before the next kick. In the smallest *Oithona davisae* and *Oithona nana* (~0.03 cm prosome length), the average $U_{min}$ was 2.8 ± 1.4 cm s$^{-1}$ (Figure 6) and increased in large (0.28–0.39 cm) species to 28.7 ± 9.7 in *Calanus helgolandicus*, 45.0 ± 15.6 cm s$^{-1}$ in *Euchirella messinensis* [60], and about 40 cm s$^{-1}$ in *Calanus finmarchicus* [22].

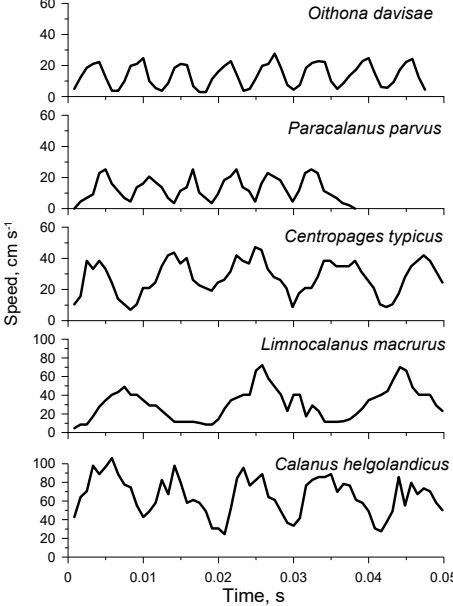

**Figure 6.** Instantaneous speeds of 5 species of copepods during the escape reaction.

It has previously been shown that both the maximum and average speed of escape reaction correlate with the size of the copepod body [20–22,28,60–63]. Our new data included the results of the video recording (1200 fps) of the escape reaction of 15 species of copepods and updated results of the old filming (3000 fps) of the escape reaction of the larger Mediterranean copepods *Euchaeta media* and *Euchirella messinensis* (Table 3).

**Table 3.** Kinematic parameters of the escape reaction in calanoid and cyclopoid copepods at 20–22 °C. $L_{pr}$: prosome length, cm; $U_{max}$: maximum instantaneous speed, cm s$^{-1}$; $U_{kick}$: mean speed of kick, cm s$^{-1}$; $D_{kick}$: total duration of kick, s; $S_{kick}$: total distance of kick, cm; $N$: number of measurements. Average values are means ± standard deviation. The literature data included in the table were obtained with a high-speed registration of at least 500 fps.

| Species | $L_{pr}$, cm | N | $U_{max}$, cm s$^{-1}$ | $U_{kick}$, cm s$^{-1}$ | $D_{kick}$, s | $S_{kick}$, cm | Source |
|---|---|---|---|---|---|---|---|
| *Oithona davisae* | 0.028 | 41 | 17.5 ± 6.3 | 10.0 ± 3.7 | 0.0081 ± 0.0023 | 0.065 ± 0.016 | |
| *Oithona nana* | 0.031 | 25 | 21.4 ± 2.5 | 10.1 ± 1.2 | 0.0076 ± 0.0009 | 0.074 ± 0.012 | |
| *Oithona similis* | 0.045 | 35 | | 12.1 ± 2.3 | 0.0077 ± 0.0011 | 0.093 ± 0.014 | |
| *Paracalanus parvus* | 0.06 | 30 | 20.8 ± 3.9 | 11.9 ± 2.5 | 0.0066 ± 0.0011 | 0.077 ± 0.013 | |
| *Pseudodiaptomus marinus* | 0.082 | 17 | 56.6 ± 7.7 | 31.9 ± 3.9 | 0.0075 ± 0.0008 | 0.238 ± 0.033 | |
| *Eurytemora affinis* | 0.08 | 13 | 38.7 ± 5.2 | 21.9 ± 2.7 | 0.0083 ± 0.0012 | 0.182 ± 0.028 | |
| *Acartia clausi* | 0.089 | 29 | 48.3 ± 9.9 | 28.1 ± 6.0 | 0.0062 ± 0.0013 | 0.170 ± 0.039 | Present data |
| *Acartia tonsa* | 0.085 | 9 | 54.5 ± 4.4 | 30.2 ± 3.2 | 0.0059 ± 0.0008 | 0.176 ± 0.022 | |
| *Centropages ponticus* | 0.084 | 5 | 27.2 ± 8.1 | 16.9 ± 4.7 | 0.0105 ± 0.0004 | 0.177 ± 0.052 | |
| *Pseudocalanus elongatus* | 0.086 | 17 | 36.0 ± 4.5 | 19.8 ± 2.8 | 0.0082 ± 0.0010 | 0.163 ± 0.037 | |
| *Centropages typicus* | 0.112 | 14 | 39.8 ± 6.1 | 22.1 ± 5.4 | 0.0120 ± 0.0031 | 0.256 ± 0.051 | |
| *Limnocalanus macrurus* | 0.19 | 18 | 55.1 ± 11.6 | 25.5 ± 4.7 | 0.0220 ± 0.0065 | 0.544 ± 0.108 | |
| *Pontella mediterranea* | 0.21 | 19 | 74.2 ± 24.6 | 44.0 ± 14.6 | 0.0112 ± 0.0025 | 0.469 ± 0.135 | |
| *Anomalocera patersoni* | 0.26 | 18 | 88.01 ± 8.9 | 57.1 ± 13.7 | 0.0095 ± 0.0014 | 0.532 ± 0.102 | |
| *Calanus helgolandicus* | 0.27 | 16 | 73.81 ± 8.3 | 45.8 ± 15.4 | 0.0150 ± 0.0050 | 0.629 ± 0.110 | |
| *Oncaea conifera* | 0.08 | 6 | | 14.7 ± 2.4 | 0.0082 ± 0.0025 | 0.204 ± 0.021 | |
| *Corycaeus limbatus* | 0.07 | 4 | | 11.3 | 0.0083 | 0.095 | |
| *Pseudocalanus elongatus* | 0.09 | 9 | 36.4 ± 6.1 | 21.2 ± 4.7 | 0.0068 ± 0.0007 | 0.142 ± 0.025 | |
| *Undinopsis similis* | 0.10 | 4 | | 9.7 ± 3.5 | 0.0137 ± 0.0027 | 0.134 ± 0.013 | |
| *Pleuromamma abdominalis* | 0.24 | 10 | | 25.0 ± 1.9 | 0.0147 ± 0.0002 | 0.386 ± 0.042 | [29] |
| *Euchaeta media* | 0.24 | 5 | | 18.3 ± 1.8 | 0.0121 ± 0.0013 | 0.220 ± 0.029 | |
| -''- | 0.29 | 3 | | 36.1 ± 2.2 | 0.0128 ± 0.0038 | 0.432 ± 0.047 | |
| *Euchirella messinensis* | 0.32 | 4 | 83.8 ± 22.0 | 41.5 ± 4.3 | 0.0153 ± 0.0008 | 0.708 ± 0.026 | |
| -''- | 0.39 | 3 | 116.0 ± 6.8 | 71.5 ± 4.5 | 0.0153 ± 0.0006 | 1.112 ± 0.105 | |
| *Anomalocera patersoni* | 0.38 | 5 | 102.9 ± 14.6 | 64.9 ± 8.3 | 0.0061 ± 0.0010 | 0.404 ± 0.108 | |
| *Oithona davisae* | 0.03 | 68 | 19.8 ± 4.2 | 10.1 ± 2.1 | 0.0074 | 0.075 ± 0.016 | |
| *Acartia tonsa* | 0.074 | 59 | 37.8 ± 9.6 | 24.1 ± 5.3 | 0.0076 | 0.185 ± 0.024 | [22,28] |
| *Calanus finmarchicus* | 0.30 | | 75.6 | | 0.013 | | |
| *Acartia tonsa* | 0.083 | 55 | 44.6 ± 15 | 25.6 ± 10 | | | |
| *Acartia lilljeborgii* | 0.103 | 56 | 48.6 ± 11.7 | 23.2 ± 7.6 | | | [19,21,58] |
| *Temora turbinata* | 0.074 | 49 | 46.3 ± 5.3 | 25.3 ± 3.3 | | | |
| *Paracalanus parvus* | 0.066 | 30 | 40.7 ± 2.9 | 22.7 ± 2.0 | | | |
| *Temora turbinata* | 0.074 | | 21.5 ± 5.5 | 10.3 ± 5.6 | | | |
| *Centropages furcatus* | 0.10 | | 20.8 ± 1.7 | 11.5 ±1.6 | | | [64] |
| *Subeucalanus pileatus* | 0.205 | | 45.3 ± 3.2 | 25.6 ± 2.5 | | | |
| *Pontella marplatensis* | 0.23 | | 47.7 ± 17.2 | 24.3 ± 9.4 | | | |
| *Parvocalanus crassirostris* | 0.039 | | 17 | | 0.0034 ± 0.004 | 0.13 ± 0.01 | [63] |
| *Eurytemora affinis* | 0.077 | | 34.2 ± 4.4 | 18.1 ± 10.2 | 0.0101 ± 0.001 | 0.21 ± 02 | |
| *Acartia hudsonica* | 0.075 | 14 | 38.7 ± 10.0 | | | | |
| *Tortanus discaudatus* | 0.122 | 21 | 53.6 ± 5.7 | | | | [20] |
| *Centropages hamatus* | 0.099 | 9 | 38.6 ± 2.8 | | | | |
| *Temoralongicornis* | 0.059 | 4 | 26.2 ± 2.8 | | | | |
| *Euchaeta elongata* | 0.41 | 8 | | 31.4 ± 4.8 | | | [65] |
| *Euchaeta rimana* | 0.24 | 7 | | 27.6 ± 3.2 | | | |
| *Paraeuchaeta elongata* | 0.40 | | 120 | | | | [66] |
| *Calanus pacificus* | 0.22 | 7 | 53 ± 7 | | | | |
| *Bestiolina similis* | 0.054 | | 26.3 ± 5.5 | | | | [67] |

These data allowed us to examine the size-scaling of escape speeds stimulated by electric impulses at 20 °C over a large size range, and $U_{max}$, $U_{min}$, and $U_{mean}$ all scaled approximately with prosome length to power of 3/4 (Figure 7).

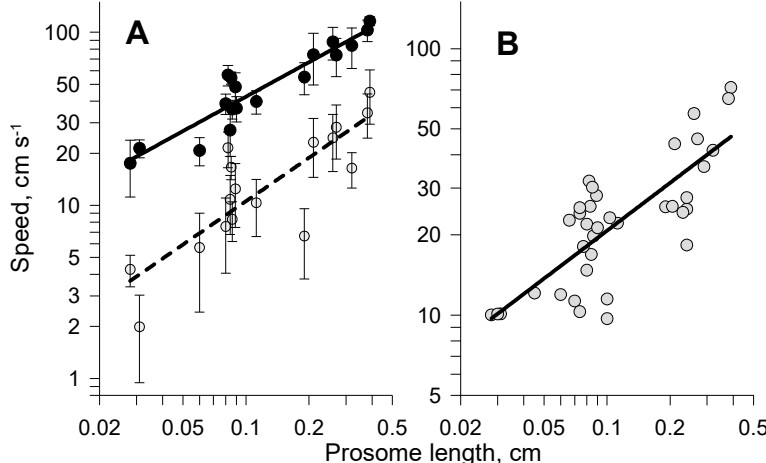

**Figure 7.** (**A**): Maximum (●) and minimum (○) species-specific instantaneous speed in a continuous sequence of kicks of the escape reaction stimulated by electrical impulses; $U_{max} = 194.9\ L^{0.66}$ ($R^2 = 0.87$) and $U_{min} = 70.0\ L^{0.83}$ ($R^2 = 0.70$). (**B**): Mean speed of escape reaction stimulated by various impulses, including predatory fish (data from Table 3); $U_{mean} = 82.0\ L^{0.60}$ ($R^2 = 0.62$).

### 3.2.2. Acceleration and Time Scale Features

Another important characteristic of the avoidance reaction is the acceleration of the body, which we calculated as $a = (U_{max} - U_{min})/t$, where $U_{max}$ and $U_{min}$ are the maximum and minimum speed during time of acceleration $t$. Body acceleration scales with size, approximately in the same way as jump speed (Figures 7A and 8A), while we did not find a significant effect of size on acceleration duration, nor on the duration of the power stroke. The total duration of a kick ($D_{kick}$), however, increased significantly with copepod size (Figure 8B).

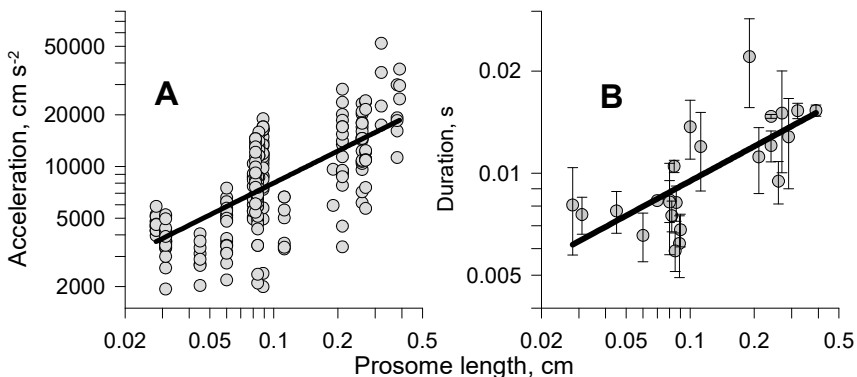

**Figure 8.** (**A**): Acceleration versus prosome length; $a \sim L^{0.62}$ ($R^2 = 0.50$). (**B**): Duration of kick; $D_{kick} = 0.021\ L^{0.34}$ ($R^2 = 0.51$).

### 3.2.3. Distance of Kicks

The number of kicks in a continuous series of escape reactions varies widely depending on the intensity and method of stimulation [21,22]. Usually, the maximum and mean speed of kicks decrease towards the end of the escape reaction due to the exhaustion of the energy resource. In addition to our old and new data, we were able to use only a few literature sources to analyze the escape movement of copepods (see Supplement Table S1). The distance covered during both the copepod stroke phase $S_{st}$ and the entire kick phase $S_{kick}$ scaled with prosome length as $L^{0.8}$ and $L^{0.88}$, respectively (Figure 9).

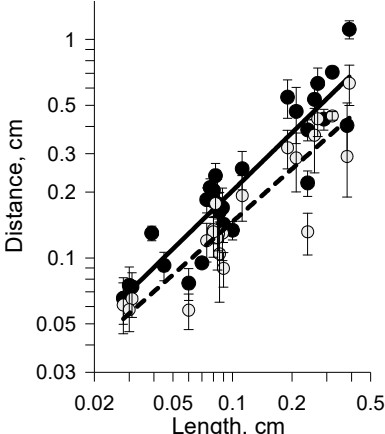

**Figure 9.** Distance covered during total kick phase (●) and stroke phase (○) approximated as $S_{\text{kick}} = 1.55\,L^{0.88}$ ($R^2 = 0.84$) and $S_{\text{st}} = 0.93\,L^{0.8}$ ($R^2 = 0.82$), respectively.

## 4. Force Estimation and Size Scaling

Forces of interest are those of drag and power stroke, and they can be determined in several ways. Drag can be directly measured by observing the sinking speed of models or immobilized specimens, or it can be measured indirectly by observing the non-propulsive deceleration of swimming specimens. The force production of beating appendages can be estimated from hydrodynamic models of the power stroke or from the equation of motion, observed velocity, and the acceleration of swimming specimens. The force production can also be directly estimated by measuring the force of hydrodynamically scaled physical models subject to a known water velocity, or it can be measured by a force sensor to which animals are attached.

### 4.1. Force Production in Copepods Tethered to Force Sensor

Comprehensive studies of the force production of copepods during cruising and jump reactions were performed using a semiconductor cantilever sensor [10,30–32,60,68] (Figure 10). The sensitivity of the sensors was sufficient to measure the force produced by the small cephalic appendages of the calanoid copepods *P. parvus* with a prosome length of 0.62 mm.

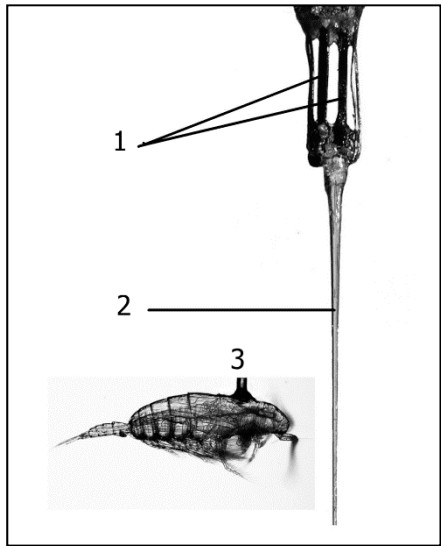

**Figure 10.** Force sensor (**1** and **2**: frontal view) and *Calanus helgolandicus* (*euxinus*) female (**3**: lateral view) attached to the end of the glass rod. **1**—four semiconductor tensoresistors of $2 \times 0.2 \times 0.05$ mm, pairwise connected in one plane according to the scheme of the Wheatstone bridge; **2**—glass rod of 4 mm length in the case of measuring the force production of copepods during the escape reaction and 8–10 mm in the case of routine locomotion.

The results are shown in Figure 11 and Table 4. Figure 11A shows the integral average (defined as the area of pulse strength divided by pulse duration) of the force production by cephalic appendages during the cruise movement ($R_{p,cr,att}$) for eight species of the Black Sea and Mediterranean copepods with a prosome lengths ($L$) from 0.062 to 0.28 mm. Despite significant differences in the kinematics of the cephalic appendages in different species, the variation of $R_{p,cr,att}$ with $L$ showed a high degree of correlation ($R^2 = 0.91$) approximated by the power-law (Figure 11A):

$$R_{p,cr,att} = 3.7\, L^{2.03}. \tag{1}$$

The same high correlation with the prosome lengths ($R^2 = 0.89$) was established for the average traction force ($R_{p,esc,att}$) of thoracic legs during escape reactions (Figure 11B):

$$R_{p,esc,att} = 384\, L^{2.2}, \tag{2}$$

as well as for the maximum instantaneous force during escape locomotion (Figure 11C; Table 3).

The average ratio of forces produced during escape and cruising locomotion has been seen to be about 100 (Table 4). This is much more than the ratio of forces during jumping and the displacement of higher aquatic and terrestrial animals, reaching only about 40 [69]. Of fundamental importance, Equations (1) and (2) show that the force production of both types of locomotion depends on the square of body size. This is consistent with M. Rubner's "surface rule", which states that in morphologically similar animals, the force available to them is proportional to the sectional area of the muscles or the square of the linear dimensions of the body [70]. Below, we consider the extent to which the length-square rule of the thrust force revealed on the attached copepods is confirmed by the kinematics and dynamics of their free swimming.

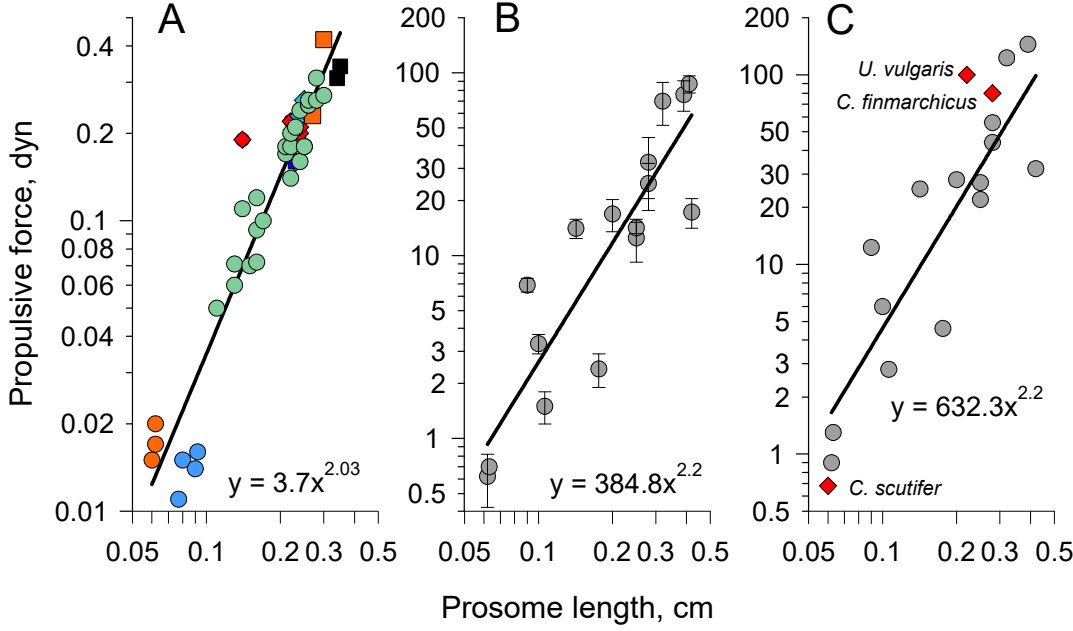

**Figure 11.** Propulsive force created by the Black and Mediterranean Seas copepods attached to semiconductor force sensor at 21 ± 2 °C (Table 3). (**A**): Mean resulting force of cephalic appendages in *Paracalanus parvus* (●), *Pseudocalanus elongatus* (●), *Calanus helgolandicus* (◌), *Phaenna spinifera* (●), *Pontella mediterranea* (◆), *Pleuromamma abdominalis* (◆), *Euchaeta marina* (■), and *Euchirella messinensis* (■) (from Svetlichny 1993a). (**B**): mean tractive force of swimming thoracic legs during the escape reaction in *Paracalanus parvus*, *Acartia clausi*, *Calanus helgolandicus*, *Pontella mediterranea*, *Undinopsis similis*, *Scolecithrix Bradyi*, *Nannocalanus minor*, *Pleuromamma abdominalis*, *Eucalanus attenuates*, and *Euchirella messinensis*. (**C**): Maximum values of the species from (**B**) and species from literature data on *Cyclops scutifer* [33], *Undinula vulgaris* [34], and *Calanus finmarchicus* [61].

**Table 4.** Propulsive forces created by cephalic limbs during cruising and by thoracic legs at escape reaction in copepods attached to force sensor. The number of individuals is shown in parenthesis.

| Species | $L_{pr}$, cm | Cruising Mean Integrated | Escape Reaction Mean Integrated | Escape Reaction Maximum Force | Source |
|---|---|---|---|---|---|
| *Paracalanus parvus* | 0.062 | 0.018 ± 0.004 (2) | 0.62 ± 0.2 (7) | 0.9 | |
| *Acartia clausi* | 0.063 | | 0.7 (2) | 1.3 | |
| -"- | 0.106 | | 1.5 ± 0.3 (4) | 2.8 | |
| *Pseudocalanus elongatus* | 0.085 | 0.014 ± 0.0022 (4) | | | |
| *Calanus helgolandicus* | 0.18 | 0.081 ± 0.02 (8) | 2.4 ± 0.5 (4) | 4.6 | |
| -"- | 0.25 | 0.019 ± 0.03 (7) | 12.5 ± 3.3 (14) | 22 | |
| -"- | 0.28 | 0.23 ± 0.04 (7) | 24.8 ± 7.1 (8) | 44 | |
| -"- | 0.28 | 0.28 ± 0.03 (3) | 32.4 ± 11.9 (12) | 56 | |
| *Pontella mediterranea* | 0.2 | 0.22 ± 0.013 (4) | 16.9 ± 3.4 (6) | 28 | [32,60] |
| *Undinopsis similis* | 0.1 | | 3.3 ± 0.4 (4) | 6 | |
| *Scolecithrix Bradyi* | 0.09 | | 6.9 ± 0.6 (4) | 12.3 | |
| *Phaenna spinifera* | 0.14 | 0.19 (1) | | | |

**Table 4.** *Cont.*

| Species | $L_{pr}$, cm | Propulsion Force, Dyn | | | Source |
|---|---|---|---|---|---|
| | | Cruising | Escape Reaction | | |
| | | Mean Integrated | Mean Integrated | Maximum Force | |
| *Nannocalanus minor* | 0.14 | | 14.1 ± 1.7 (3) | 25 | |
| *Pleuromamma abdominalis* | 0.25 | 0.22 ± 0.04 (4) | 14.2 ± 1 (11) | 27 | |
| *Eucalanus attenuatus* | 0.42 | | 17.3 ± 3.2 (6) | 32 | |
| *Euchaeta marina* | 0.32 | 0.37 ± 0.08 (2) | | | |
| *Euchirella messinensis* | 0.32 | 0.34 (1) | 70 ± 18.4 (4) | 123 | |
| -"- | 0.39 | | 76 ± 14.4 (3) | 145 | |
| -"- | 0.41 | | 87 ± 9.3 (4) | 159 | |
| *Cyclops scutifer* | 0.06 | | | 0.68 | [33] |
| *Undinula vulgaris* | 0.22 | | | 125 | [34] |
| *Calanus finmarchicus* | 0.28 | | | 80 | [61] |

## 4.2. Drag on Falling Models and Specimens

The first task in the study of force production in freely moving copepods was to determine the drag on the body. Often, results for geometrically simple bodies are used as an approximation: a sphere (e.g., [22]) or an ellipsoid of revolution simulating the body of calanoid copepods *Paracalanus* and *Centropages* without protruding organs [71]. Here, we estimated drag coefficients, $C_{d.}$, on carved wooden scale models that passively descended in a viscous fluid with different body orientations and antennae positions [72,73] (Figure 12). The hydrodynamic drag coefficient $C_d$ was determined from the defining relation:

$$C_d = 2R_d/\rho S U^2, \tag{3}$$

where the drag force $R_d$ equals the submerged body weight (dyn), $\rho$ (g cm$^{-3}$) is the density of the liquid, $S$ (cm$^2$) is the sectional area (taken to be the area of a circle with a diameter $d$ equal to the width of the prosome), and $U$ is the observed sinking speed. $C_d$ depends on the Reynolds number, Re = $\rho L U/\mu$, where $\mu$ denotes the dynamic viscosity. To compensate for the enlarged scale in these experiments, viscosity was adjusted by using glycerin–water mixtures (hydrodynamic scaling).

Later, the same principle was applied to immobilized individuals of 17 species of copepods [74]. After immobilization, the copepods were weighed in water on a modified Salvioni balance to determine their submerged body weight, and the rate of passive sinking was determined. To expand the range of Reynolds numbers, microparticles of lead were inserted into the body cavity. The drag coefficients of the body calculated from Equation (3) on the basis of the weight and speed of passive sinking are presented in Figure 12.

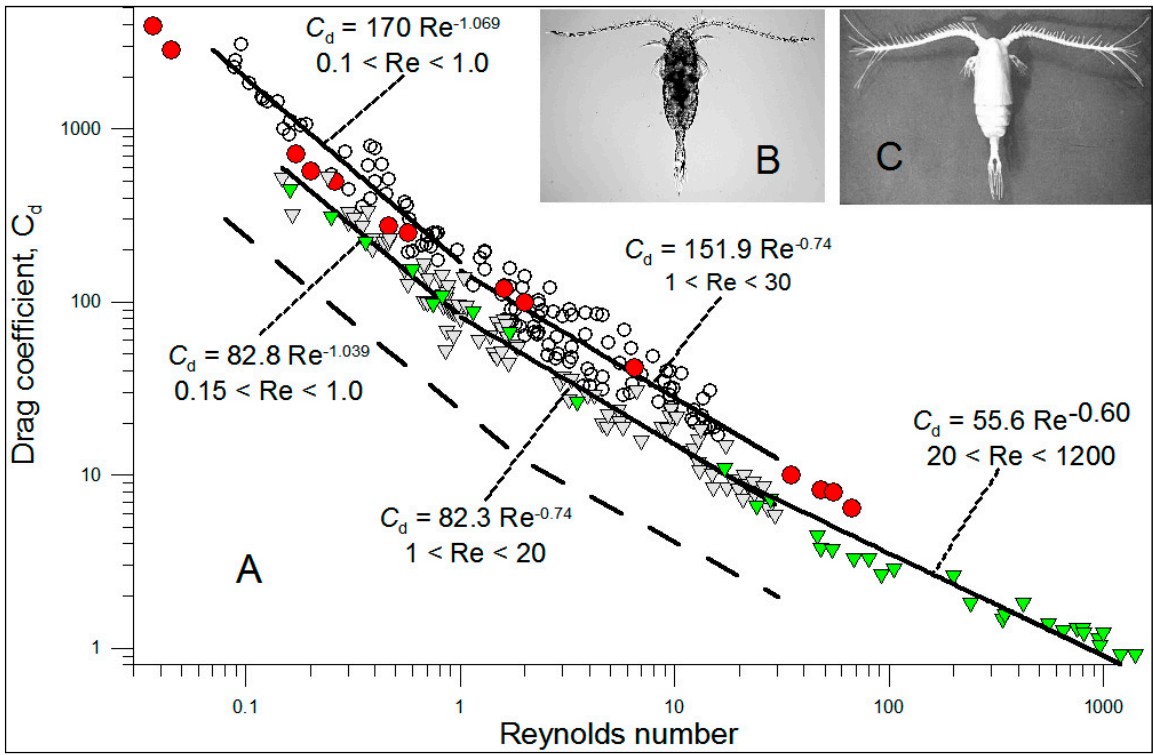

**Figure 12.** (**A**): Drag coefficient versus Reynolds number of 17 species of immobilized copepods (from [74]) when moving in water with open antennae (○) and antennae pressed to the body (▽). Red circles and green triangles indicate the $C_d$ of enlarged models when moving with spread and pressed antennas, respectively (from [72,73]). The dashed line shows $C_d$ of the sphere [75]. (**B,C**): Photos of immobilized calanoid copepod *Paracalanus parvus* and its enlarged (~1:100) model, respectively.

In Figure 12, two groups of data are distinguished: the case of movement with spread antennas, which is typical for slow cruise swimming, and the case of movement with folded antennas, which is typical for jumping movement. In general, the data turned out to be close to those obtained on enlarged models (Figure 12). To simplify the relationship, $C_d \sim f\{Re\}$ was approximated in each range of the Re scale by the relation [76]:

$$C_d = c\, Re^{-n}, \tag{4}$$

where $c$ is the hydrodynamic shape factor and $Re = d\,U/\nu$, where $\nu$ is the kinematic viscosity, $cm^2\,s^{-1}$, and $d$ is body diameter (cm) corresponding to the largest width of the prosome. The estimated coefficients $c$ for the different Re ranges (0.1–30.0 and 0.15–1200 Re for cruising and jumping, respectively) are shown in the correlation equations in Figure 12. Below, we use the experimentally determined drag coefficients to estimate force production from observed swimming speed and acceleration.

### 4.3. Detailed Analytical Model of Cruising Locomotion

At steady rectilinear translational motion, the drag of the body $R_d$ equals the resulting propulsive force $R_p$ created by the limbs in a time-averaged sense:

$$R_d = R_p, \tag{5}$$

If we multiply Equation (5) by the body velocity, the power $N_d = R_d\,U$ is the energy dissipation by drag, which equals the power effectively transferred to maintain the motion: $N_p = R_p\,U_{legs}$. However, the power actually expended by the limbs is much greater: $N_{legs} \gg N_d = N_p$, where $N_{legs}$ is the total power of action of all cephalic limbs: second antennae, mandibles, maxillae, and maxillipeds

(in *Calanus*, for example, with type of feeding; Figure 1A), because not all expended power by legs results in thrust.

To determine the power actually expended by the limbs, detailed measurements of the force and speed of individual cephalic limbs of attached cruising *Calanus helgolandicus* were carried out [10]. By determining the individual force production by second antennae, mandibles, maxillae, and maxillipeds after removing all other pairs of head limbs, it was found that the sum of these individual force productions added up to three times the force production of an intact specimen. Hence, the total power of all beating legs in this species can be estimated from the empirical relation:

$$N_{\text{legs,att}} = 3\ R_{\text{p}}\ U_{\text{a}}, \tag{6}$$

$$U_{\text{a}} = 2\ \pi\ (\alpha/180)\ F\ l_{\text{a}}, \tag{7}$$

where $U_{\text{a}}$ is the circular speed of the second antenna relative to body, $F$ denotes frequency of beat, $l_{\text{a}}$ is the second antenna length measured from the point of attachment to the body to the middle of the length of the end bristles (Figure 13), and $\alpha$ is the angular amplitude of the legs rotation that varies near 50° for feeding current feeders like *P. elongatus*, amounts to 80–90° for cruising feeders, like *C. helgolandicus*, and amounts to 100–120° for pontellids species (our personal observations based on high speed video; see Figure 1).

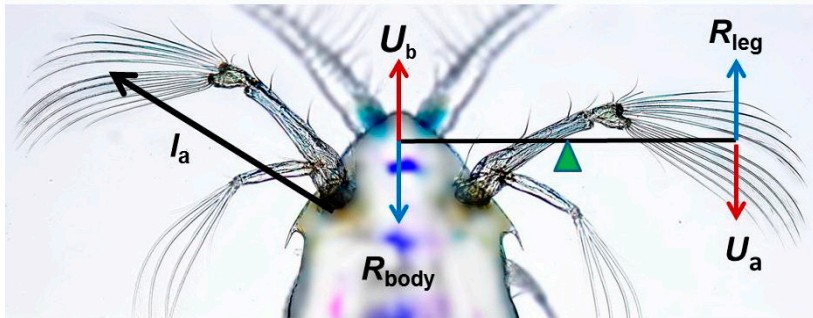

**Figure 13.** Second antennas of *Pontella mediterranea*. The black arrow on the left shows the length of the antennae $l_{\text{a}}$; other arrows show the forces and speeds of the body and legs. The magnitudes of velocities $U_{\text{b}}$ (= $U$) and $U_{\text{a}}$ determine the location of a simulated rowlock (green triangle) of an oar model.

In free swimming copepods, however, the beating legs act on water with the effective velocity of ($U_a$–$U$), and, by taking into account the empirical value $k$ (possibly different from 3) of the hydrodynamic efficiency of locomotion, the total power of all beating legs of the free cruising copepods can be determined as:

$$N_{\text{tot}} = R_{\text{d}}\ U + k\ R_{\text{p}}\ (U_a - U), \tag{8}$$

The first term in Equation (8) is the power of thrust transferred to the body for it to overcome the drag (i.e., the useful thrust power), while the second empirical term represents the extra power dissipated by the moving limbs, a quantity that is not useful for propulsion. The drag force on the body is calculated based on the average speeds for each of the studied species (Table 2) from the usual equation of drag expressed in terms of an empirical drag coefficient $C_{\text{d}}$ (recall Equation (3)):

$$R_{\text{d}} = \frac{1}{2}\ C_{\text{d}}\ \rho\ S_{\text{body}}\ U^2, \tag{9}$$

Taking $S = \pi d^2/4$ and $C_{\text{d}}$ from Equation (4) with n = 0.74 for 1 < Re < 30 (see Figure 12), we obtain:

$$R_{\text{d}} = 59.7\ \nu^{0.74}\ \rho_{\text{w}}\ d^{1.26}\ U^{1.26}, \tag{10}$$

Using the data of Table 2 for cruising copepods led to the scaling $R_d = 11.5\,L^{2.82}$ ($R^2 = 0.86$) (Figure 14), which differed from the scaling $R_{p,att} \sim L^{2.03}$ for attached copepods (Figure 11).

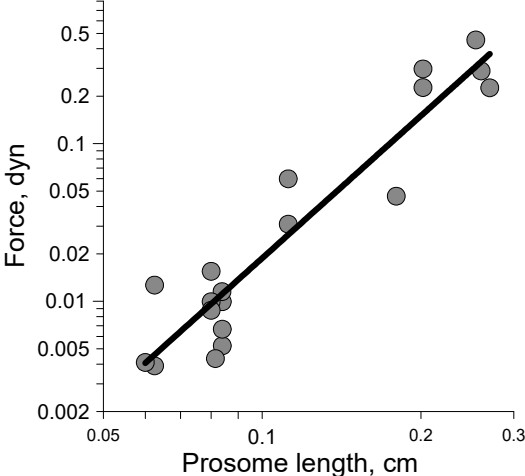

**Figure 14.** Drag force versus prosome length calculated for free cruise-swimming copepods, ($R_d = 11.5$ $L^{2.82}$ ($R^2 = 0.86$). All points are average values.

In the smallest tethered calanoid copepods *Paracalanus parvus* and *Pseudocalanus elongatus*, $R_{p.att}$ was significantly ($p < 0.001$) 2.5 times higher than $R_d$ ($0.017 \pm 0.005$ and $0.0068 \pm 0.004$ dyn, respectively), whereas in the largest species, there was no difference because of different values of the empirical factor $k$ in Equation (8), as shown in [10] and seen from Figure 14, probably due to the higher hydrodynamic efficiency of the paddle locomotion at higher Reynolds numbers.

It has previously been shown that the flow field around tethered copepods differs from that around a grazing free-swimming animal [20,27,56,77]. However, the difference in the scaling of force production output and available force to overcome body drag may also be due to a change in the hydrodynamic efficiency of the type of locomotion (coefficient $k$ in Equation (8)). For this reason, the muscle force realized by attached individuals approximately scales as $R_{p,att} \sim L^2$, but it scales as $R_d \sim L^3$ in freely cruising copepods (Figure 14). Taking into account that the same species were used in our experiments with attached and free copepods, we could test this hypothesis by calculating the propulsive force $R_p$ of freely moving individuals as the drag force on beating limbs using the following equation:

$$R_p = \frac{1}{2}C_{d,leg,att}\,\rho_w\,S_{leg}\,(U_a - U)^2, \tag{11}$$

where $U_a$ is the circular speed of the second antenna (Equation (7)), $S_{leg}$ is the cross sectional area of legs, and $C_{d,leg,att}$ is the drag coefficient of attached individuals calculated as:

$$C_{d,leg,att} = 2\,R_{p,cr,att}/\rho_w\,S_{leg}\,U_a^{\,2}, \tag{12}$$

From all measurements, we found the correlations $C_{d,leg,att} = 34.5\,Re_l^{-0.88}$ ($R^2 = 0.91$), where $Re_l = U_a\,l_a/\nu$ and $R_p = 5.46\,L^{2.36}$ (Figure 15). This may indicate that the propulsive force of the limbs, directly measured in tethered copepods or predicted for free-swimming individuals, is more consistent with the scale $L^2$.

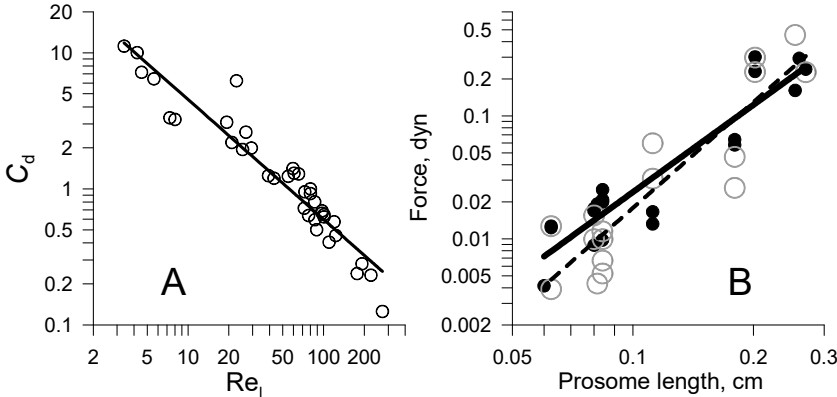

**Figure 15.** (**A**): Coefficient of hydrodynamic resistance $C_{d,leg,att}$ ($\circ$) from Equation (12) for cephalic limbs of 8 copepod species attached to a force sensor. (**B**): Propulsion force $R_p$ ($\bullet$) calculated from experimental data for the same free cruise-swimming copepods based on $C_{d,leg,att}$ and, for comparison, the values of drag force $R_d$ ($\circ$, dotted line) calculated by Equation (10).

Next, we calculated the power required to overcome body drag and resistance of cephalic limbs' actions, the two terms $N_d = R_d U$ and $N_p = k (U_a − U)R_p$ in Equation (8). The results in Figure 16 indicate that power that is sufficient overcome body drag scales as $N_d \sim L^{4.1}$, while for limbs, it scales as $N_p \sim L^{3.1}$ or as $\sim M^{1.0}$, where $M$ denotes body mass. A similar regression coefficient ($L^{3.04}$) was obtained when calculating the power of attached cruising copepods using equation $N_{p,att} = k U_a R_{p,att}$.

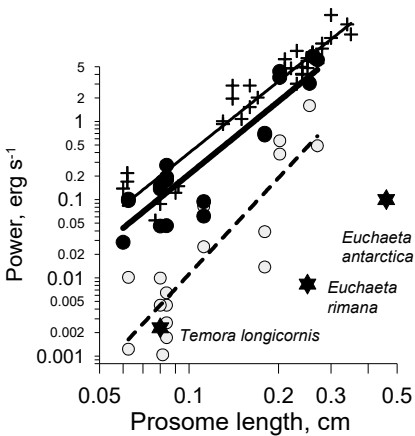

**Figure 16.** Calculated power versus prosome length required for free cruise-swimming copepods to overcome body drag ($\circ$, thin dotted line, $N_d = 137.7 L^{4.09}$) and to move cephalic limbs ($\bullet$, solid line, $N_p = 266.07 L^{3.1}$), as well as the results for attached individuals (+, thin line, $N_{p,att} = 436.57 L^{3.04}$). Black asterisks indicate literature data for *Temora longicornis* [17]; *Euchaeta rimana* [26], *and Euchaeta Antarctica* [27].

To calculate $N_p$, we used the empirical value from Equation (6), $k = 3$, for *C. helgolandicus* [10]. However, when we took into account the difference in the type of cephalic appendages action and the efficiency of locomotion of small cruising feeder copepods compared to large cruising feeders and especially pontellids, in which the cephalic appendages do not oppose each other during the creation of propulsive force, the slope of the regression line became less than 3.0. In other words, the scaling $R_p \sim L^{3.0}$ can lead to an underestimation of the power consumption of small species and an overestimation in large ones. Such a correction corresponds to the scaling of the energy potential of animals [78] whose biological power is usually proportional to $M^{0.67–1.0}$.

### 4.4. Analytical Model of Escape Reaction

One way to obtain estimates of forces and energy change during an escape jump from measured kinematics is to use the equation of motion of body mass $M$ during acceleration $dU/dt$ due to propulsive force $R_p$ and opposing drag $R_d$:

$$M \, dU/dt + R_d = R_p, \tag{13}$$

First, for non-propulsive deceleration, Equation (13) provides an estimate of the drag force as a function of velocity:

$$R_d = -M \, dU/dt, \tag{14}$$

However, such estimates prove to be quite inaccurate because they depend on the numerical discretization of the time derivative of second order of position. Therefore, it is more accurate to assume the validity of measured relations for the drag coefficient of sinking specimens (Figure 12) and calculate the drag force from the usual relation Equation (9):

$$R_d = \frac{1}{2} \, C_d \, \rho_w \, (\pi/4) \, d^2 \, U^2, \tag{15}$$

where $C_d = 55.6 \, Re^{-0.60}$ for the range of $10 < Re < 1200$ (Figure 12), which corresponds to our studied copepods with a body width of $d = 0.013$–$0.13$ cm and a mean speed $U = 10$–$100$ cm s$^{-1}$ at constant temperature of 20 °C (Table 3, Supplement Table S1).

The values of $R_d$ calculated by Equation (15) using the average speed of cyclopoid and calanoid copepods during the stroke phase of escape reactions increased on the average from 0.1 dyn in small oithonids to 30 dyn in the largest calanoid copepods (Figure 17A) according to the scaling $R_d \sim L_p^{2.15}$. Using this relation and observed accelerations in Equation (13) led to the scaling $R_p \sim L_p^{2.55}$. A similar procedure for attached calanoid copepods gave the close scaling $R_{p\,att} \sim L_p^{2.37}$ (Figure 17B).

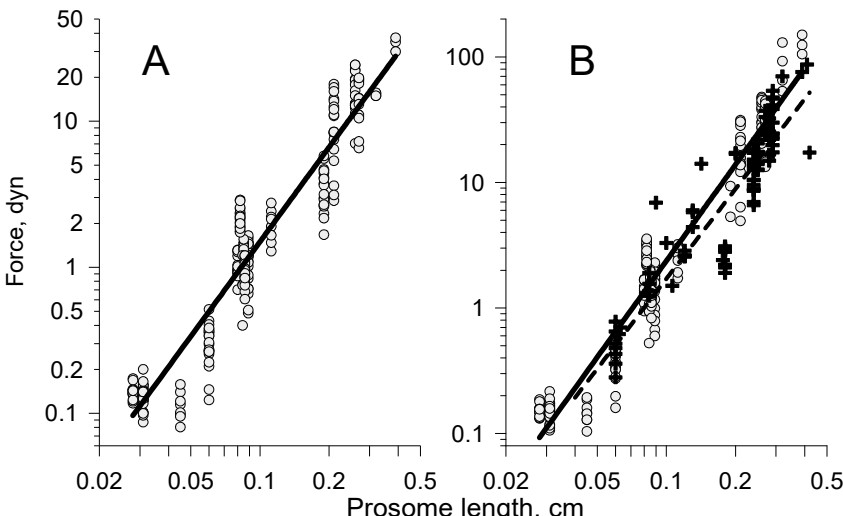

**Figure 17.** (**A**): Drag force; $R_d = 211.6 \, L^{2.15}$ ($R^2 = 0.91$, N = 241). (**B**): Propulsive force, calculated ($\circ$) for free swimming copepods and directly measured (+) in attached individuals during an escape reaction; $R_p = 838.2 \, L^{2.55}$ ($R^2 = 0.94$) and $R_{p,att} = 408.6 \, L^{2.37}$ ($R^2 = 0.82$, N = 88).

Multiplying the equation of motion (Equation (13)) by the velocity of the body during kick stroke phases and integrating it over the time of acceleration during which the velocity increases from $U_{min}$ to $U_{max}$, the energy expended ($\Delta E_{stroke}$) is obtained as:

$$M \, \frac{1}{2} \, (U_{max}^2 - U_{min}^2)_1 + <U \, R_d \, \Delta t >_1 = <U \, R_p \, \Delta t >_1 \equiv \Delta E_{stroke}, \tag{16}$$

where $< >_1$ signifies an integrated quantity over time interval $\Delta t_1$.

Following the power stroke phase, the limbs are retuned back during time $\Delta t_2$, while the velocity decreases from $U_{\max}$ in the end of stroke phase back to $U_{\min}$, for which the energy balance gives:

$$M \frac{1}{2}(U_{\max}{}^2 - U_{\min}{}^2)_2 - <U\,R_d\,\Delta t>_2 = \Delta E_{\text{limb,back}}, \tag{17}$$

which merely shows that deceleration is caused by body and limb drag.

Including the so-called 'energy-leg-back' contribution, the total energy expended by limbs during all kick stroke phases is $E_{\text{sum}} = \Delta E_{\text{stroke}} + \Delta E_{\text{limb,back}}$, or:

$$E_{\text{sum}} = M \frac{1}{2}(U_{\max}{}^2 - U_{\min}{}^2)_1 + <U\,R_d\,\Delta t>_1 + M \frac{1}{2}(U_{\max}{}^2 - U_{\min}{}^2)_2 - <U\,R_d\,\Delta t>_2, \tag{18}$$

Using Equation (18), the mean power of an escape kick $N_{\text{esc}}$, defined as $N_{\text{esc}} = E_{\text{sum}}/D_{\text{kick}}$ (where the duration is $D_{\text{kick}} = \Delta t_1 + \Delta t_2$) was calculated to vary in the range from 1 to 4000 erg s$^{-1}$ following the scaling $N_{\text{esc}} \sim L^{3.05}$ (Figure 18). This result turned out to be very close to the power of attached calanoids that scale as $N_{\text{esc,att}} \sim L^{2.99}$, which was calculated as $N_{\text{esc,att}} = R_{p,\text{att}}\,U_{\text{leg}}$, where $U_{\text{leg}} = 2\,\pi$ $(\alpha/180)\,F\,h_a$, $\alpha = 145°$, and $h_a = 0.75\,l_a$ according [23]. In both cases, $N_{\text{esc}}$ was seen to scale linearly with body mass $M$.

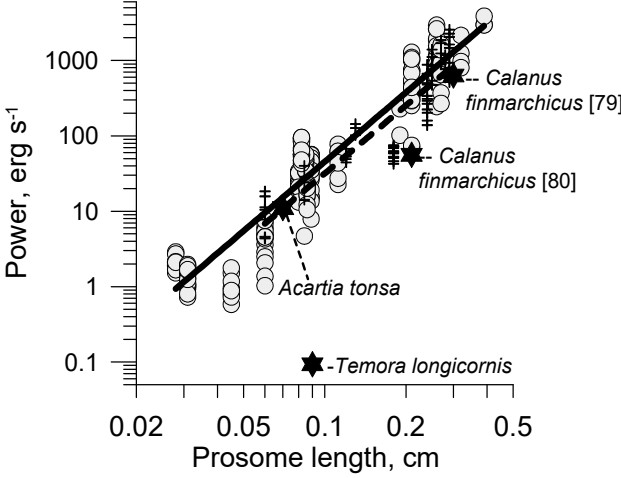

**Figure 18.** Power of kick $N_{\text{esc}}$ during the escape reaction of free swimming ($\bigcirc$) and attached (+) copepods. The power regressions were $N_{\text{esc}} = 51000\,L^{3.05}$ ($R^2 = 0.91$, N = 190) and $N_{\text{esc,att}} = 31400L^{2.99}$ ($R^2 = 0.82$, N = 71), respectively. Black asterisks indicate literature data for *Temora longicornis* [25]; *Acartia tonsa* and *Calanus finmarchicus* from [79], and *C. finmarchicus* according [80].

## 5. Discussion and Conclusions

The cruising speeds of calanoid copepods vary widely depending on the type of feeding and the associated mechanism of creating propulsive force, on the body density, and on the water temperature. The density of the body is significantly higher than the density of water [45]. In this regard, the speed of passive sinking can distort the real speeds that are provided by the movement of the limbs. For many calanoid copepods, the available cruising speed is only two-to-three times higher than the speed of gravitational sinking [45,81,82]. For example, in females of *C. helgolandicus* at 20 °C, sinking speed can reach 0.8 cm s$^{-1}$ [83]. Therefore, at the maximum swimming speed available to them (see Table 1), their speed changes 2.5 times from 1.4 to 3 cm s$^{-1}$, depending on the direction of movement being down or up. Temperature affects speed through changes in the viscosity and density of water [84], but it changes to a greater extent due to changes in the rate of muscle contraction. The rate of many biological systems, including planktonic crustaceans [35], varies in proportion to the temperature coefficient $Q_{10} = 2$, i.e., an increase of a factor 2 when the temperature increases by 10 °C. This has

been confirmed in experiments examining the temperature response of limb beat frequency and the swimming speed of copepods [54,82,85]. Therefore, we used video recordings of horizontal cruise swimming calanoid copepods from the Black, Marmara, and Baltic Seas at the same temperature 20 °C, as well as literature data for swimming copepods at similar temperatures.

### 5.1. Scaling of Kinematic and Mechanical Parameters of Cruising

The average cruising speed and cephalic limb beat frequency scaled as $U = 13.4\ L^{1.4}$ (Table 2) and $F = 16.0\ L^{-0.53}$, respectively. According to the reviews [86,87] that investigated the scale laws of mechanics and kinematics of "biological motors" of different systematic groups, such empirical slopes of $U$ and $F$ correspond to cyclic motors with mass $M > \sim 0.4$ mg (fruit fly size and above) whose maximum force output scales as $R \sim L^{3.0}$ or $R \sim M^{1.0}$.

In our analysis of free swimming cruising copepods, we found a scaling of the body drag force $R_d \sim L_{pr}^{2.82}$ similar to that of cyclic motors (Table 5), whereas in tethered copepods, the measured force production scaled as $R_{p\ cr,att} \sim L_{pr}^{2.06}$ or $M^{0.69}$. According to [86], animals whose maximum force output scales as $M^{0.67}$ correspond to a group of steady translational (i.e., linear) motors. However, Marden [86] noted that: "there are potentially many force outputs by translational motors .... that fall between the two fundamental scaling relationships ... " $R \sim L^{2.0}$ and $R \sim L^{3.0}$.

**Table 5.** Exponents $m$ in scaling relations versus body length, $L^m$.

| Quantity and Condition | Cruising | | | | Escape Jump | | | |
| --- | --- | --- | --- | --- | --- | --- | --- | --- |
| | Free Swimming | | Attached Locomotion | | Free Swimming | | Attached Locomotion | |
| | $m$ | Figure | $m$ | Figure | $m$ | Figure | $m$ | Figure |
| Body speed, $U$ | 1.4 | Figure 4 | | | 0.7 | Figure 7 | | |
| Drag force, $R_d$ | 2.82 | Figure 14 | | | 2.15 | Figure 17A | | |
| Propulsive force, $R_p$ | 2.36 | Figure 15 | 2.06 | Figure 11A | 2.55 | Figure 17B | 2.2 | Figure 11B |
| Power | 3.1 | Figure 16 | 3.04 | Figure 16 | 3.05 | Figure 18 | 2.94 | Figure 18 |

Note that the above difference in scaling of $R_d$ and $R_{p,cr,att}$ revealed by us was mainly due to smaller values of $R_d$ in small species (see Figure 14), the magnitude of which can be illustrated as follows. In order for the predicted $R_d$ of smallest free swimming *P. parvus* to increase to the level of $R_{p,cr,att}$ in the attached individuals of this species, their average speed should be two times higher than our measured speeds (see Table 1). Hence, scaling according to $L^{2.0}$ may be the best estimate for all sizes. The total cruising power $N_{cr}$ of copepods in the size range $0.06 < L < 0.3$ cm, calculated on the basis of the force and speed of the cephalic appendages, varied on average from 0.05 to 5 erg s$^{-1}$ (or from 0.05 to $5 \times 10^{-7}$ W) in proportion to $L^{3.1}$ or $\sim M^{1.0}$ (Figure 16). This is consistent with the scaling of metabolic energy available for the long-term cruising of animals, which usually scales as $M^{0.67-1.0}$ [78], while the net power needed to move the body, calculated based on body drag $N_d$ and speed, has an excessively high exponent $L^{4.1}$ or $M^{1.4}$ (Figure 16). According to our estimation, the efficiency of locomotion defined as $N_d/N_{sum}$ changed, on average, from 5% in *P. parvus* up to 20% in pontellids.

Few other studies have dealt with the mechanical power of cruise swimming copepods, and all of these have calculated the rate of energy dissipation in the liquid volume due to the movement of the limbs of a cruising copepod. For only one species, an adult female *Temora longicornis* [17], the power ($2.3 \times 10^{-10}$ W) was close to our estimated power to overcome the body drag in copepods of the same size ($L \sim 0.08$ cm, about $3 \times 10^{-10}$ W, Figure 16). In two other similar studies, the energy dissipation by *Euchaeta* rimana [26], and especially *Euchaeta antarctica* [27], turned out to be almost two orders of magnitude smaller than for copepods of the similar size from our experiments. The discrepancy can be partially explained by the fact that this very large Antarctic copepod swam in cold water (0 °C) at a speed (1.5 cm s$^{-1}$) that was approximately three times lower than the expected speed at 20 °C in a copepod of the same size (Figure 4A). Similarly, the speed of subtropical *E. rimana* at 20 °C (0.7 cm s$^{-1}$) [57] was three times lower than that of *C. helgolandicus* of the same size.

## 5.2. Scaling of Kinematic and Mechanical Parameters of Escape Reaction

The escape reaction for all copepods is carried out by a simple sequence of strokes with morphologically similar thoracic swimming legs {37} and, apparently, with similar efficiency. Therefore, the predicted correlations of $R_d$ and $R_p$ for free swimming and $R_p$ measured in tethered copepods during escape reaction were more consistent with each other than in the case of cruising (Table 3).

The observed scaling of escape speeds with body size, $U_{mean} \sim L^{0.7}$ and $U_{max} \sim L^{0.66}$, as well as drag and force production (Table 3), are more consistent with the translational motors whose maximum force output scales as $L^{m<3.0}$ [86]. Indeed, the measured propulsive force of copepods attached to the force sensor scaled as $R_{p,att} \sim L^{2.15}$, and the calculated forces of free escapes scaled as $R_{d,free} \sim L_p^{2.36}$ and $R_{p,free} \sim L^{2.55}$. The average values of $R_{p,att}$ for the smallest calanoid copepod *P. parvus* (0.62 ± 0.2 dyn), as well as for the largest *E. messinensis* (87 ± 9 dyn), did not differ significantly from the calculated values of $R_{p,free}$ (Figure 17B).

Nevertheless, the total power of free copepods during the escape reaction turned out to scale as $N_{esc} \sim L^{3.06}$ and for the attached as $N_{esc} \sim L^{2.94}$. Thus, the total power of both free and attached copepods during the escape reaction turned out to scale as $L^{3.0}$. This trend in $N_{esc}$ was confirmed by the results of calculations by Jiang and Kiørboe [79], who estimated the maximum values of mechanical power for *Acartia tonsa* (0.069 cm prosome length) and *Calanus finmarchicus* with a prosome length of 0.3 cm as $1.1 \times 10^{-6}$ and $6.3 \times 10^{-5}$ W, respectively. Muphy et al. [80] determined the value of maximum power delivered to the fluid by the swimming legs of *C. finmarchicus* (*L* = 0.21 cm) to be $5.6 \times 10^{-6}$ W. The maximum energy delivered by swimming appendages defined by Duren and Videler [25] in *Temora longicornis* (*L* = 0.09 cm) equaled $9.3 \times 10^{-9}$ W. This was almost two orders less, probably due to the relatively low $U_{max}$ of the studied individuals (10.8 cm s$^{-1}$) in comparison with the $U_{max}$ of the escape reaction of this species stimulated by hydrodynamic stimuli (26.2 cm s$^{-1}$) [20].

## 5.3. Cost of Transport during Cruising and Jumping

In general, the propulsive force and the power created by the swimming limbs are two and three, respectively, orders of magnitude higher than the force and power created by the head appendages. However, it is more correct to assess the differences between the two types of swimming of the copepods by the energy costs of transport ($C_t$) [78], defined as the energy consumption per unit of body mass and distance travelled ($S$): $C_t = E/M\,S = N/M\,U$ (cal g$^{-1}$ km$^{-1}$).

For large calanoid copepods (*L* = 0.2–0.3 cm), the average mechanical cost of transport moving by unsteady jumps ($C_{tm}$ = 45.2 ± 15 cal g$^{-1}$ km$^{-1}$) is seven times higher than by steady cruise swimming ($C_{tm}$ = 6.7 ± 3.1 cal g$^{-1}$ km$^{-1}$), while for small calanoid copepods (*L* = 0.06 cm), it is only about three times higher ($C_{tm}$ = 74.4 ± 24 and $C_{tm}$ = 25.1 ± 16 cal g$^{-1}$ km$^{-1}$, respectively) and the ratio is even less in the smallest copepods (Figure 19A). Thus, for large copepods, the cost of transportation is much higher for swimming-by-jumping than for cruise swimming, while for small ones, the difference is not so large. There are advantages to swimming-by-jumping, the first being hydrodynamic stealth: swimming-by-jumping creates only a relatively small fluid disturbance and, thus, is less susceptible to rheotactic predators than copepods that cruise steadily [28]. This may explain why only small copepods (cyclopoids) swim by jumps, while larger copepods are cruise swimmers.

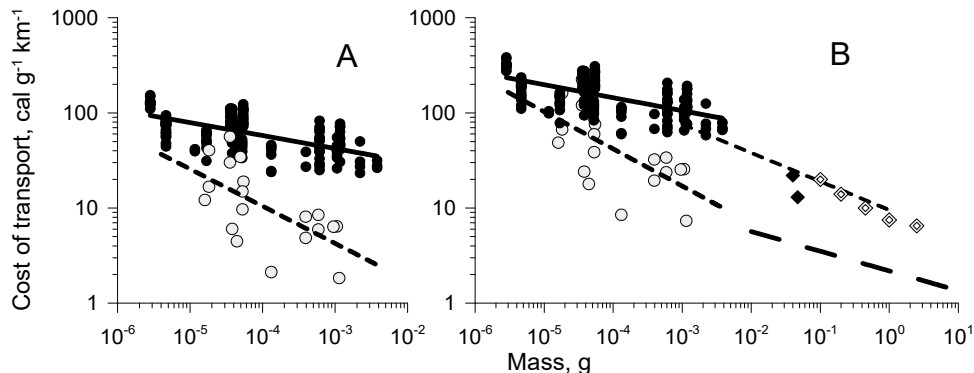

**Figure 19.** (**A**): Maximum mechanical cost of transport ($C_{tm}$) during escape reaction (•) and cruising (○). (**B**): Metabolic cost of transport ($C_{tb}$) and values for swimming fish (long dashed line) and flight of insects (short dashed line) [78]; escape reaction of shrimp (unshaded diamond) [88] and *Euphausia* (black diamond) [89].

The biological cost of transport $C_{tb}$ is due not only to the mechanical efficiency of locomotion but also to the efficiency of muscle contraction. The theoretical maximum efficiency of muscle contraction efficiency is 0.5 [90]. However, with prolonged cruise work, the maximum coefficient of mechano-muscular efficiency of aerobic muscles does not exceed 0.25. With short-term muscle action during the escape reaction, it can increase to 0.4 [91]. To compare our measurements with observations for other species recorded in the literature, we multiplied our estimates of the mechanic costs of transportation by factors of 4 and 2.5 for cruising and escape jumping, respectively (Figure 19B). Transportation costs for escape jumps were found to be in line with those for other arthropods [23,24], and cruise swimming was found to be consistent with swimming costs in fish (not startle responses).

**Supplementary Materials:** The following is available online at http://www.mdpi.com/2311-5521/5/2/68/s1, Table S1: Kinematic parameters of the escape reaction in calanoid and cyclopoid copepods at 20–22 °C.

**Author Contributions:** L.S. conceived, designed, and performed the experiments; L.S., P.S.L., and T.K. analyzed the data; all authors contributed to writing and have approved the final version of the manuscript. All authors have read and agreed to the published version of the manuscript.

**Funding:** The Center for Ocean Life is supported by the Villum Foundation. We further acknowledge support from the Danish Council for independent Research (7014-00033B).

**Acknowledgments:** This work was supported by the projects of the NASU (grant number 0114U002041). The experimental part of this study was carried during 1984–2016 in the Department of animal physiology of Institute of Biology of the Southern Seas, Sevastopol, Ukraine, Faculty of Aquatic Sciences of Istanbul University (2007–2019), Turkey and in the SYKE Marine Research Center (2019), Finland.

**Conflicts of Interest:** The authors declare no conflicts of interest.

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
