# Peer review of "Kinematic and Dynamic Scaling of Copepod Swimming"

_fluids, doi:10.3390/fluids5020068_

Round 1
Reviewer 1 Report
This manuscript reviews the kinematics of copepods during cruising and jumps. In addition, it gathers data on speed, force, and cost of transport of copepods from theirs and other sources and provides a regression as a function of size. In general, I like the manuscript and such synthesis of data is important. Nevertheless, I have some concerns regarding the analysis and the analytical model provided.
General comments:
- All the analysis and the regressions are done on dimensional parameters except the drag coefficient for Fig. 12. I am wondering why not all the parameters of non-dimensionalized. Non-dimensionalization reduces the number of parameters and improves the fit. Consider the the example of the drag coefficient vs. Re in Fig. 12, if it was plotted as drag vs length then for each velocity a different curve had to be plotted or the spread would have been wider. Note that drag not only depends on size but also velocity. Similarly, the propulsive force (thrust) also depends on both velocity and size. Consequently, plotting it as a force coefficient vs Reynolds will provide a better fit (higher R^2). Speed also depends on size and frequency, so plotting it as Re vs. Strouhal number is more appropriate. Finally, plotting power coefficient (P/0.5 rho U^3 L^2) as a function of Re makes more sense.
- The limitations are not discussed, e.g., what is the effect of more data on the goodness of fits, or if R squared can be improved by some other type of fit (other than linear regression). The mains assumptions such as the linear correlation between the parameters and size are not discussed.
Specific comments:
Ln 114: minimal?
Ln 170, 620: citation should be numbered for Lewis et al.
Ln 341: Do you mean drag coefficient? For sinking at constant speed Drag=Weight – Buoyancy
Ln 436: mention why the energy of expended by legs is more than the energy for locomotion, i.e., because it also move the water in other directions (than just pushing them backward).
Ln 441-444: It is implicitly assumed that the force generated by each appendage is independent of the others. This is not necessarily true as the motion of one appendage can change the flow for others, i.e., affect the force production by others. The synergy is ignored.
That might be the reason for the coefficient 3. The summation of the force produced by each appendage should be equal to the total force by the appendages.
Ln 479-482: the statements are not clear to me. It is not clear why the coefficient k might be different. The difference in force due to free swimming vs tethered makes more sense. In fact, when free swimming the drag coefficient can be different than tethered due to the difference in Re (Cd is a function of Re).
Ln 525-526: Why estimating forces based on acceleration is not accurate? A justification is needed.
Author Response
Dear Editor,
dear Reviewer,
we wish to thank you for the useful and detailed comments on our manuscript. Please find below a point-by-point reply to the comments raised in the first round of review. All issues were addressed in details, so as to make the revised version of the manuscript as complete as required by the Reviewer.
We hope the present document matches integrally with the expectations of the Reviewer.
Comments and Suggestions for Authors
This manuscript reviews the kinematics of copepods during cruising and jumps. In addition, it gathers data on speed, force, and cost of transport of copepods from theirs and other sources and provides a regression as a function of size. In general, I like the manuscript and such synthesis of data is important. Nevertheless, I have some concerns regarding the analysis and the analytical model provided.
General comments:
- All the analysis and the regressions are done on dimensional parameters except the drag coefficient for Fig. 12. I am wondering why not all the parameters of non-dimensionalized. Non-dimensionalization reduces the number of parameters and improves the fit. Consider the the example of the drag coefficient vs. Re in Fig. 12, if it was plotted as drag vs length then for each velocity a different curve had to be plotted or the spread would have been wider. Note that drag not only depends on size but also velocity.
- Similarly, the propulsive force (thrust) also depends on both velocity and size. Consequently, plotting it as a force coefficient vs Reynolds will provide a better fit (higher R^2). Speed also depends on size and frequency, so plotting it as Re vs. Strouhal number is more appropriate. Finally, plotting power coefficient (P/0.5 rho U^3 L^2) as a function of Re makes more sense.
- Reply : The objective of our study has been to determine the scaling by size (prosome length) of a number of key parameters of copepod swimming. The suggested classical engineering approach of using dimensionless numbers is well taken and would be possible if complete data sets were available, which they unfortunately are not. Furthermore, if the mentioned force coefficient were presented as function of Reynolds number (and perhaps also Strouhal number), as well as Re versus St, it would not be possible for the reader to extract the separate size effect of say force or velocity. More relations are needed.
The limitations are not discussed, e.g., what is the effect of more data on the goodness of fits, or if R squared can be improved by some other type of fit (other than linear regression). The mains assumptions such as the linear correlation between the parameters and size are not discussed.
-Reply: The basis for all scaling relations used in this study has been power-law regression because it is most suitable to data sets covering a range of several decades. It is also the standard way of exploring allometric relations in biology, be it metabolic rate, growth rates, or swimming speed, etc. This is also the approach taken in Marden (2005), a key work on the topic.
Specific comments:
Ln 114: minimal?
-Reply: yes, now changed
Ln 170, 620: citation should be numbered for Lewis et al. –
-Reply: yes, thank you, now done
Ln 341: Do you mean drag coefficient? For sinking at constant speed Drag=Weight – Buoyancy.
-Reply: this is an opening – or introductory – statement announcing what follows, so it should not include details. We modified this phrase in line 335 as follows: “Drag can be measured directly by observing the sinking speed of models or immobilized specimens or indirectly by observing non-propulsive deceleration of swimming specimens.”.
Ln 436: mention why the energy of expended by legs is more than the energy for locomotion, i.e., because it also move the water in other directions (than just pushing them backward).
-Reply: yes, no propulsive device would have a 100% mechanical efficiency. The explanation now given in line 429 reads: “….type of feeding, Fig. 1A), because not all expended power by legs results in thrust.“ We complete the new line 432-433 with the phrase: “By determining the individual force production by second antennae, mandibules, maxillae and maxillipeds after removing all other pairs of head limbs it was found that the sum of these individual force productions added up to 3 times the force production of an intact specimen”.
Ln 441-444: It is implicitly assumed that the force generated by each appendage is independent of the others. This is not necessarily true as the motion of one appendage can change the flow for others, i.e., affect the force production by others. The synergy is ignored.
-Reply: Yes, there undoubtedly will be some interaction between beating limbs due to the flows generated, but the approach was used to accurately determine both speed and force generated by each set of limbs. If all were beating simultaneously it was difficult or not possible to assign an appropriate speed to recorded force during the event.
That might be the reason for the coefficient 3. The summation of the force produced by each appendage should be equal to the total force by the appendages.
- Reply: With the simultaneous multidirectional action of the limbs, their resulting force is 3 times less, while the total power is 3 times greater than that which corresponds to the product of the propulsion force and the speed of the second antennae.
Ln 479-482: the statements are not clear to me. It is not clear why the coefficient k might be different. The difference in force due to free swimming vs tethered makes more sense. In fact, when free swimming the drag coefficient can be different than tethered due to the difference in Re (Cd is a function of Re).
- Reply: We complete the line 467 with the phrase “probably due to the higher hydrodynamic efficiency of paddle locomotion at higher Reynolds numbers”, suggesting that for small current feeders at lower Reynolds numbers, the viscous interaction between limbs and surface of the body leads to a decrease in the output useful for movement by propulsive forces, in comparison with large cruising feeders at a large Reynolds numbers.
Ln 525-526: Why estimating forces based on acceleration is not accurate? A justification is needed.
Reply: Text in line 515-516 changed to read: “….to be quite inaccurate because they depend on the numerical discretization of the time derivative of second order of position. Therefore it…..”

Reviewer 2 Report
Summary
In this study, the authors use both previously published and newly collected data to perform scaling analysis on a variety of copepod species of different sizes swimming in their two characteristic locomotion modes: cruising and jumping. They scale kinematic parameters including swimming speed, beat frequency, and acceleration as well as dynamic parameters including force generation, drag coefficient, power, and cost of transport. The kinematic scaling section incorporates new kinematics data gathered using high speed filming from a variety of species performing cruise swimming and escape swimming. The dynamic scaling section uses previously acquired data gathered using a force sensor and copepods with intact legs or various legs removed. The dynamic scaling section also develops a novel analytical model of the forces generated by the body and the legs during untethered cruise swimming in order to show how scaling of tethered cruising copepods differs from that of free-swimming cruising copepods. A similar analytical model of the forces on escaping copepods is developed. The cost of transport scaling gives insight into why small copepods swim using a lower energy hopping mode while large copepods typically swim via cruising.
This is a solid study that compiles decades worth of data to provide extensive scaling relationships. I recommend its publication after a few minor revisions.
Major Comments
-The authors provide a good review of previous literature on copepod swimming, especially the early work, which was very interesting.
-The description of body pitching in section 3.2 is interesting. It would be worth citing Kiørboe et al. (2010) in this context since they measured body pitching up to 55 deg in a jumping Acartia tonsa.
Kiørboe, T., Andersen, A., Langlois, V. J. and Jakobsen, H. H. (2010). Unsteady motion: Escape jumps in planktonic copepods, their kinematics and energetics. J. R. Soc. Interface 7, 1591-1602.
-In Figure 12, it is not clear which equation goes with which fitted line
-In lines 425-429, it is not clear which ranges of Re you are talking about. Please define these ranges.
-In the paragraph beginning on line 439, it is not clear whether you are removing all but one pair of appendages and then repeating this with all but a different set of appendages removed.
-On line 466, you refer to using the data of Table 3 for cruising copepods, but I think you mean to refer to Table 2.
-What are the other thinner lines in Figure 14? These are not referenced.
-In general, Section 4.3, is not very clear. Can you please try to clarify the text here?
-In Section 5.1, you discuss the mechanical power of cruise swimming and reference a few studies which have measured this power and plot these on Figure 16. There is a good match for Temora longicornis but not for Euchaeta rimana or Euchaeta antarctica. The poor fit for both Euchaeta species seems to be attributed to the cold water temperature, but E. rimana is a subtropical species. It would be good to clarify this and to speculate on why the difference exists for this species.
-Also in Section 5.1, you reference one work by van Duren but not the following one, which has more measurements of power requirements of cruising in Temora:
van Duren, L. A., Stamhuis, E. J., & Videler, J. J. (2003). Copepod feeding currents: flow patterns, filtration rates and energetics. Journal of Experimental Biology, 206(2), 255-267.
How do these measurements stack up against your model?
-In Section 5.2 and Figure 18, you similarly discuss the mechanical power of escape jumping but do not compare your model with available measurements. There are two that I know of that could be compared.
First, the van Duren et al (2003) paper referenced above has measurements for tethered Temora longicornis.
Second, the following paper by Murphy et al (2012) has measurements of work and power of a free-swimming escaping Calanus finmarchicus.
Murphy, D. W., Webster, D. R., & Yen, J. (2012). A high‐speed tomographic PIV system for measuring zooplanktonic flow. Limnology and Oceanography: Methods, 10(12), 1096-1112.
It would be very interesting to include these measurements in Figure 18 as you do for cruise swimming Figure 16.
Minor Comments
-On line 502, I don’t think M has been defined previously. I assume this is mass.
-There are a number of typos, some of which I have outlined below by line number:
- 121: “the cephalic appendages”
- 207: negatively
- In Table 2, it is not clear why (-“-) is used in the lower rows next to several species names.
- In Table 3, commas are used in place of decimal points in some of the rows
- The x axis of Figure 9 is mis-spelled
- 366: kinematics
- 399: “descend in viscous fluids”
- 422: jumping
- 547: “in the stroke phase”?
- 599: use decimal instead of comma in exponent
Author Response
Dear Editor,
dear Reviewer,
we wish to thank you for the useful and detailed comments on our manuscript. Please find below a point-by-point reply to the comments raised in the first round of review. All issues were addressed in details, so as to make the revised version of the manuscript as complete as required by the Reviewer.
We hope the present document matches integrally with the expectations of the Reviewer.
Comments and Suggestions for Authors
Summary
In this study, the authors use both previously published and newly collected data to perform scaling analysis on a variety of copepod species of different sizes swimming in their two characteristic locomotion modes: cruising and jumping. They scale kinematic parameters including swimming speed, beat frequency, and acceleration as well as dynamic parameters including force generation, drag coefficient, power, and cost of transport. The kinematic scaling section incorporates new kinematics data gathered using high speed filming from a variety of species performing cruise swimming and escape swimming. The dynamic scaling section uses previously acquired data gathered using a force sensor and copepods with intact legs or various legs removed. The dynamic scaling section also develops a novel analytical model of the forces generated by the body and the legs during untethered cruise swimming in order to show how scaling of tethered cruising copepods differs from that of free-swimming cruising copepods. A similar analytical model of the forces on escaping copepods is developed. The cost of transport scaling gives insight into why small copepods swim using a lower energy hopping mode while large copepods typically swim via cruising.
This is a solid study that compiles decades worth of data to provide extensive scaling relationships. I recommend its publication after a few minor revisions.
Major Comments
- The authors provide a good review of previous literature on copepod swimming, especially the early work, which was very interesting. Reply: Thank you
- The description of body pitching in section 3.2 is interesting. It would be worth citing Kiørboe et al. (2010) in this context since they measured body pitching up to 55 deg in a jumping Acartia tonsa. Kiørboe, T., Andersen, A., Langlois, V. J. and Jakobsen, H. H. (2010). Unsteady motion: Escape jumps in planktonic copepods, their kinematics and energetics. J. R. Soc. Interface 7, 1591-1602.
-Reply: line 286 is supplemented by the text: Nevertheless in small Acartia tonsa (<0.1cm) body angle can vary within 55° [13].
- In Figure 12, it is not clear which equation goes with which fitted line.
-Reply: Yes. Dashed lines from correlation equation to a point on the relevant linear section of regression lines have now been added.
- In lines 425-429, it is not clear which ranges of Re you are talking about. Please define these ranges.
Reply: The sentence “The estimated coefficients for the different Re ranges are shown in Fig. 12.” has been changed to read: “The estimated coefficients c for the different Re ranges (0.1 – 30.0 Re and 0.15 – 1200 Re for cruising and jumping, respectively) are shown in the correlation equations in Fig. 12.” Moreover the studied ranges of Re for two types of motion are indicated.
- In the paragraph beginning on line 439, it is not clear whether you are removing all but one pair of appendages and then repeating this with all but a different set of appendages removed.
Reply: All but one pair of limbs were removed for each measurement. line 432 gives the explanation: "By determining the individual force production by second antennae, mandibles, maxillae and maxillipeds after removing all other pairs of head limbs it was found.”
- On line 466, you refer to using the data of Table 3 for cruising copepods, but I think you mean to refer to Table 2.
Reply: Yes, it is now corrected to Table 2.
- What are the other thinner lines in Figure 14? These are not referenced.
Reply: Yes, are now named in the figure caption by the added sentence: “Thin lines indicate the 95% confidence intervals for each fitted regression.”
- In general, Section 4.3, is not very clear. Can you please try to clarify the text here?
Reply: We have struggled with this section for the original submission to make it as clear as possible. It is admittedly difficult to read, but that owes partly to the derivations being convoluted.
- In Section 5.1, you discuss the mechanical power of cruise swimming and reference a few studies which have measured this power and plot these on Figure 16. There is a good match for Temora longicornisbut not for Euchaeta rimana or Euchaeta antarctica. The poor fit for both Euchaeta species seems to be attributed to the cold water temperature, but rimana is a subtropical species. It would be good to clarify this and to speculate on why the difference exists for this species.
Reply: The poor fit for both Euchaeta species is due to the very low swimming speed. We supplemented the text with this information (new Line 609): “Similarly, the speed of subtropical E. rimana at 20°C (0.7 cm s-1) [57] was 3 times lower than that of C. helgolandicus of the same size.”
- Also in Section 5.1, you reference one work by van Duren but not the following one, which has more measurements of power requirements of cruising in Temora: van Duren, L. A., Stamhuis, E. J., & Videler, J. J. (2003). Copepod feeding currents: flow patterns, filtration rates and energetics. Journal of Experimental Biology, 206(2), 255-267.
How do these measurements stack up against your model?
Reply: You are right, this is a citation error. We changed the citation by adding the correct one to the list of references.
- In Section 5.2 and Figure 18, you similarly discuss the mechanical power of escape jumping but do not compare your model with available measurements. There are two that I know of that could be compared.
First, the van Duren et al (2003) paper referenced above has measurements for tethered Temora longicornis.
Second, the following paper by Murphy et al (2012) has measurements of work and power of a free-swimming escaping Calanus finmarchicus.
Murphy, D. W., Webster, D. R., & Yen, J. (2012). A high‐speed tomographic PIV system for measuring zooplanktonic flow. Limnology and Oceanography: Methods, 10(12), 1096-1112.
It would be very interesting to include these measurements in Figure 18 as you do for cruise swimming Figure 16.
Reply: - Thank you for your comment. We supplemented Figure 18 with data from van Duren & Videler (2003), Murphy et al (2012) and Jiang & Kiørboe (2011), and also analyzed these data in section 5.2 with new text in Lines 625 – 633.
Minor Comments
-On line 502, I don’t think M has been defined previously. I assume this is mass. Reply: Now changed to read (line 494) «…or as ~M1.0, where M denotes body mass.” . (see also Table 1. List of symbols)
-There are a number of typos, some of which I have outlined below by line number:
- 121: “the cephalic appendages”
- 207: negatively
- In Table 2, it is not clear why (-“-) is used in the lower rows next to several species names.
- In Table 3, commas are used in place of decimal points in some of the rows Reply: -Indicates additional data on same specimen size
- The x axis of Figure 9 is mis-spelled
- 366: kinematics
- 399: “descend in viscous fluids”
- 422: jumping
- 547: “in the stroke phase”?
- 599: use decimal instead of comma in exponent
Reply: -Thank you. All typos corrected, including commas-to-periods in Table S1
Submission Date
30 March 2020
Date of this review
16 Apr 2020 18:14:23

Round 2
Reviewer 1 Report
The authors have adequately responded to my comments. My main comment about using dimensionless parameters were not implemented due to unavailability of all conditions from different studies.
Author Response
Dear Reviewer,
We want to thank you for the helpful and detailed comments on our manuscript raised in the first round of the review. All questions were examined in detail to make the revised version of the manuscript as complete as the reviewer required.
We hope that this document is fully consistent with the expectations of the reviewer.
Reviewer 2 Report
The authors have sufficiently improved the manuscript so that I think it is ready for publication.
Author Response

(The authors gave the same response as above.)
